# STAR: Speculative Decoding with Searchable Drafting and Target-Aware Refinement for Multimodal Generation

## Abstract

Speculative decoding (SD) has proven to be an effective technique for accelerating autoregressive generation in large language models (LLMs), however its application to vision-language models (VLMs) remains relatively unexplored. We propose *STAR*, a novel SD framework designed specifically for fast and efficient decoding in VLMs. STAR leverages a neural architecture search (NAS) framework with target-aware supernet training to automatically identify both the optimal interaction strategy between the draft and target models, and the most suitable draft model architecture for the underlying hardware implementation platform. STAR additionally incorporates adaptive intermediate feature distillation, guided by attention entropy, to enable efficient draft training. Experiments on a range of well-established VLMs, including LLaVA series, Pixtral, and SmolVLM, demonstrate that STAR achieves up to a $3.8\times$ speedup compared to standard decoding approaches and significantly outperforms existing SD baselines in both inference throughput and speculative acceptance length across a wide spectrum of VLMs.

## 1 Introduction

Vision-Language Models (VLMs) play a pivotal role in advancing artificial intelligence by integrating visual perception with natural language understanding. These models empower machines to process and generate both visual and textual data, enabling a broad array of applications such as image captioning (Zhou et al., 2020; Hu et al., 2022; Chen et al., 2022; Dzabraev et al., 2024), visual question answering (Chappuis et al., 2022; Bazi et al., 2023; Wang et al., 2024), and content-based search (Li et al., 2024b; Sun et al., 2025). Despite their impressive capabilities, VLMs are computationally demanding, primarily due to the complexity of integrating high-dimensional visual and textual inputs. Speculative decoding (SD) (Stern et al., 2018; Leviathan et al., 2023) accelerates the autoregressive generation process of large language model (LLM) by dividing it into two stages: a low-cost drafting phase and a parallel verification phase. This allows multiple candidate tokens to be generated and then verified simultaneously in a single forward pass through the target LLM. The approach boosts decoding efficiency while maintaining output quality through a selective acceptance-rejection mechanism.

As highlighted in prior work (Chen et al., 2023; Li et al., 2024c;d; Cai et al., 2024; Ankner et al., 2024; Xia et al., 2023a; Zhang et al., 2023; Miao et al., 2023; Chen et al., 2024b; Hu et al., 2025), achieving superior performance in the SD framework requires the draft model to meet two key criteria. First, it should achieve a high acceptance ratio, meaning that most of its proposed tokens are validated by the target model. Second, it should deliver low execution latency to minimize overall decoding time. Balancing these factors is essential for navigating the accuracy–latency trade-off, a challenge well-suited to neural architecture search (NAS), which has been widely demonstrated to yield highly effective trade-offs in similar settings.

Although speculative decoding techniques have been widely developed to accelerate inference in LLMs, their integration into multimodal language models (Li et al., 2024a; Raj et al., 2024), especially VLMs, has received relatively little attention. In this paper, we introduce *Speculative Decoding with Searchable Drafting and Target-Aware Refinement for Multimodal Generation* (STAR). Specifically, STAR employs a neural architecture search (NAS) mechanism with target-guided dis-

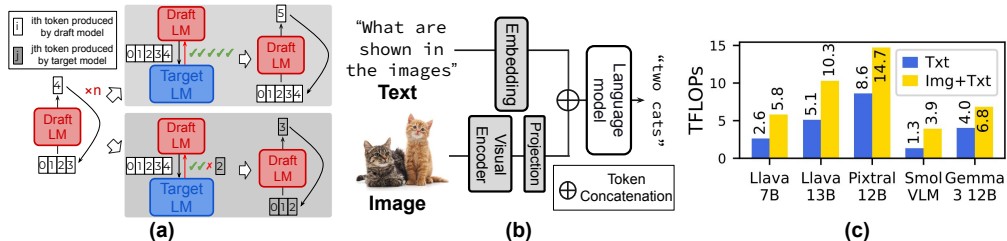

Figure 1: (a) Speculative decoding process, LM denotes large model. (b) Architecture of Vision-language model (c) Computational cost in Tera FLOPs of VLMs processing text only (Txt) and multi-modal (Img+Txt) inputs over different VLMs.

tillation to train a supernet encompassing diverse draft configurations, then identify the optimal draft model configuration, input pruning ratio, and interaction strategy with the target model, all tailored to the underlying hardware platform. Another key innovation of STAR is its selective utilization of intermediate-layer representations, which capture the most informative features from both modalities. These representations serve as effective supervision signals, enabling the draft model to achieve high accuracy. While neural architecture search and model pruning are well-established optimization techniques, their systematic application to draft model design in VLM speculative decoding remains unexplored. The key challenge lies not in developing new algorithms, but in formulating this multimodal acceleration problem within a principled search framework and designing VLM-specific optimization dimensions.

We evaluate STAR on a diverse set of widely used vision-language models, including LLaVA-v1.6-Vicuna-7B/13B (Liu et al., 2023a), SmolVLM-2B (Marafioti et al., 2025), and Pixtral-12B (Agrawal et al., 2024), across a range of multimodal tasks. Extensive experiments show that STAR significantly outperforms established speculative decoding baselines, while maintaining high acceptance rates across various applications. Our main contributions are as follows:

- STAR integrates a NAS framework to identify the optimal draft model configuration for optimal speedup. The search process further determines the optimal input and model pruning ratio and the most efficient connection strategy between the draft and target models for the optimal speedup performance.

- During training, STAR dynamically selects intermediate features from the target model's middle layers based on proposed criteria, using them to supervise the draft model, which improves its predictive accuracy and extends token acceptance lengths. Additionally, STAR employs a cross-attention mechanism to leverage these intermediate outputs, enabling more effective knowledge transfer from the target model and resulting in significant performance improvements.

- Evaluation results demonstrate that STAR is able to achieve up to a $3.8\times$ speedup compared to conventional decoding methods across various VLMs and tasks, surpassing existing speculative decoding approaches. **The code used for implementation is available in the supplementary materials**.

## 2 RELATED WORK

**Speculative decoding.** is an effective approach to alleviating the sequential bottleneck in language model inference (Stern et al., 2018). It divides the decoding process into two stages: a lightweight *draft model* quickly generates a sequence of candidate tokens, which are then verified in parallel by a more accurate *target model*, as illustrated in Figure 1(a).

Let the draft model $M_{da}$ generate $\gamma$ draft tokens $(t_1, \ldots, t_\gamma)$ in each draft step. During the verification phase, the target model $M_{ta}$ evaluates these tokens in parallel, but accepts them sequentially. If all tokens in the batch are accepted, the draft model proceeds to generate the next set of candidate tokens. (upper branch of Figure 1(a)). Otherwise, the target model supplies the correct token and assists the draft model in generating subsequent tokens (lower branch of Figure 1(a)). Specifically, it checks whether each draft token $t_i$ matches the output of its own sampling. If a mismatch occurs

at position $i$, all tokens from $t_i$ onward are discarded, and the target model's sampled token at position $i$, denoted $t_i'$, is used instead. The accepted token sequence is therefore $(t_1, \ldots, t_{i-1}, t_i')$. SD allows for parallel token generation, moving beyond the conventional step-by-step decoding, while the verification phase ensures output quality by accepting or rejecting the draft tokens.

**Vision-Language Models.** VLMs are designed to jointly process visual and textual inputs, enabling machines to interpret and generate content that integrates both modalities. As shown in Figure 1(b), a typical VLM consists of a *visual encoder* and a *language model*. The image is first processed by the visual encoder to extract visual tokens, which are then concatenated with textual tokens and passed into the language model to produce the final output. More recent models like LLaVA (Liu et al., 2023b), InstructBLIP (Dai et al., 2023), and Pixtral (Agrawal et al., 2024) focus on improving zero shot generalization by aligning model responses with human intent through instruction tuning. While large VLMs achieve strong performance, their high computational cost and memory usage pose challenges for deployment on devices with limited resources. To address this issue, lightweight VLMs such as TinyGPT-V (Yuan et al., 2023) and TinyLLaVA (Zhou et al., 2024) aim to build more efficient architectures. SmolVLM (Marafioti et al., 2025) expands this direction by introducing a family of compact models with parameter sizes ranging from 256 million to 2 billion, achieving competitive results with significantly reduced model size.

To quantify the computational cost introduced by visual input processing, we measure the FLOPs required by several models, including LLaVA-v1.6-Vicuna-7B, Pixtral-12B, and SmolVLM-2B, and Gemma3-12B (Gemma Team et al., 2025), using the ScienceQA dataset. We select a representative example that includes a $480 \times 300$ image, a prompt of 166 tokens, and a generated output of 500 tokens. FLOPs are computed using the PyTorch Profiler. As shown in Figure 1(c), processing both image and text inputs results in an average increase of $2.1 \times$ in computation compared to text-only inputs, highlighting the importance of developing more efficient visual processing methods. This additional cost does not only come from the one-time visual encoder prefill, but mainly from the autoregressive decoding stage: all visual tokens are stored in the KV cache and participate in attention at every decoding step, so each generated token must attend to both text and image tokens. Over a full sequence, this repeated interaction with a large number of visual tokens dominates the end-to-end FLOPs increase that we observe. In the rest of the paper, STAR is designed to target exactly this bottleneck by compressing visual tokens and pruning attention in the draft model, thereby reducing decoding cost while preserving a high token acceptance rate.

**Neural Architecture Search.** By algorithmically exploring the vast architecture space, NAS alleviates the time-consuming process of training models with different configurations to find the most efficient and effective designs for specific tasks. Traditional NAS methods can be categories into two classes. The first class of methods searches directly for the optimal architecture by making the search process differentiable. These approaches (Liu et al., 2018; Wan et al., 2020; Chen et al., 2019) formulate architecture search as a differentiable optimization problem. They apply continuous relaxation to express each operation as a weighted combination of candidate operations, allowing architecture parameters to be optimized jointly with network weights using gradient-based methods. The second class of methods jointly trains a collection of nested neural networks and then employs a dedicated search network to select the optimal architecture from the trained candidates. Once For All (OFA) (Cai et al., 2019; Chen et al., 2020; Cai et al., 2018; Zhang et al., 2022) first trains a *SuperNet* containing diverse architectural configurations across four dimensions (depth, width, kernel size, and resolution), then applies progressive shrinking to train from large to small sub-networks. It uses a trained neural network and hardware-specific lookup tables to predict optimal sub-networks given target hardware and constraints such as latency budgets. Unlike conventional NAS approaches that train supernets in isolation, STAR employs target-aware supernet training, where the draft model supernet is trained using intermediate features and distillation signals from the target model. This allows the search process to optimize not only for computational efficiency but also for alignment with the target model's internal feature representations, leading to a better speedup ratio.

## 3 METHODOLOGY

The training and inference workflows of STAR are illustrated in Figure 2. During training, the draft model processes input tokens containing both visual and textual modalities, and is optimized

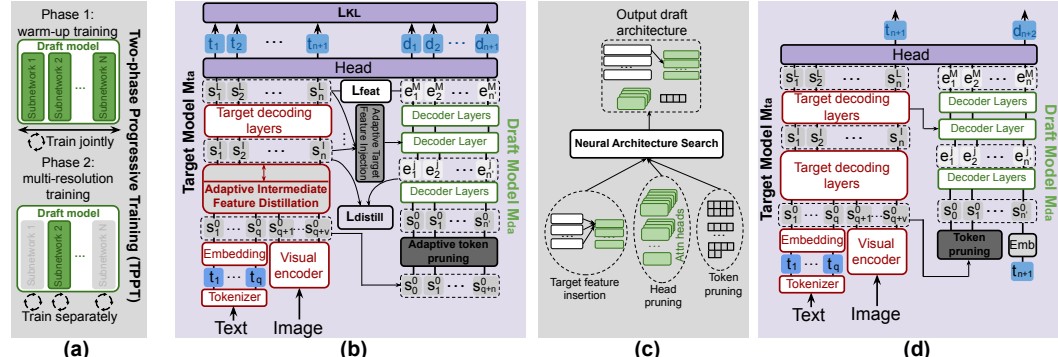

Figure 2: STAR framework overview. (a) Two-Phase Training: supernet training followed by sub-network sampling. (b) Training with three losses: $\mathcal{L}_{KL}$, $\mathcal{L}_{feat}$, and $\mathcal{L}_{distill}$. (c) NAS exploring head pruning, token compression, and feature injection. (d) Inference operation of STAR.

using the proposed *two-phase progressive training* (TPPT) strategy outlined in Section 3.1. TPPT includes two stages: *warm-up training* and *multi-resolution training*. During inference, as shown in Figure 2(d), the draft model with the highest speedup on the target hardware platform is selected to optimize performance. Furthermore, the draft model's configuration can be dynamically adjusted in response to varying hardware conditions. STAR leverages intermediate features from the target model through two distinct mechanisms. First, during inference, a cross-attention layer in the draft model dynamically incorporates features from a target layer selected through neural architecture search, providing real-time guidance for token generation. Second, during training, we introduce an adaptive intermediate layer distillation (AIFD) approach (Section 3.2), where a separate adaptive distillation loss uses features from an optimally chosen target layer to supervise draft model learning.

### 3.1 TWO-PHASE PROGRESSIVE TRAINING FOR DRAFTING

For the simplicity of interpretation, let $M_{ta}$ and $M_{da}$ denote the target and draft models, containing $L$ and $M$ transformer blocks, respectively. Let $t_n$ and $d_n$ be the $n$-th tokens generated by $M_{ta}$ and $M_{da}$. For the target model $M_{ta}$, the input consists of $q$ text prompt tokens and $v$ visual tokens. We define $s_n^{j-1}$ and $e_n^{j-1}$ as the intermediate hidden state of the $n$-th token at layer $j$ in $M_{ta}$ and $M_{da}$, respectively, with a total of $H_{j-1}$ attention heads. Figure 2(a) illustrates the Two-Phase Progressive Training (TPPT) procedure, which consists of two main stages. In the first, warm-up training phase, the entire draft model is trained. This includes the entire set of weights within $M_{da}$.

The training process is illustrated in detail in Figure 2(b). At the initial decoding stage, the target model receives both textual and visual prompt tokens $t_1, \ldots, t_n$, where $n = q + v$, and begins predicting the subsequent token $t_{n+1}$. The draft model $M_{da}$ then predicts the $(n + 2)$-th token, denoted as $d_{n+2}$. To enhance the quality of draft token generation, we integrate intermediate features from both models using a cross-attention mechanism. Specifically, we extract features from a selected layer $l$ of the target model $M_{ta}$, repre-

Table 1: Impact of head pruning and input compression on speculative decoding performance.

| Configuration | MMT-Bench | |
|---|---|---|
| | S | $\tau$ |
| Vanilla | 2.45 | 6.50 |
| Head pruning (0.75) | 2.57 | 6.31 |
| Input compression (0.7) | 2.56 | 6.40 |

sented as $S^l = (s_1^l, s_2^l, \ldots, s_n^l)$, and from the $j$-th layer of the draft model $M_{da}$, represented as $E^j = (e_1^j, e_2^j, \ldots, e_{n'}^j)$, where $n' = q + r$, and $0 \leq r \leq v$ is the number of visual tokens fed to the draft model. In this cross-attention setup, $E^j$ serves as the query, while $S^l$ provides the keys and values.

The second phase, *multi-resolution training*, utilizes the OFA framework to train draft subnetworks. Our NAS framework operates across three dimensions grounded in established transformer optimization principles, and we validate the necessity of searching each dimension through preliminary

experiments on LLaVA-v1.6-Vicuna-7B. First, attention head pruning leverages observations that transformer attention heads exhibit varying importance for model performance (Michel et al., 2019; Voita et al., 2019; Xia et al., 2023b). To test this, we apply magnitude-based pruning to remove 25% of the heads and observe a 6% improvement in speedup, as shown in Table 1. Second, visual token compression exploits the fact that visual tokens contribute unequally to final predictions in vision-language models (Chen et al., 2024a; Shang et al., 2024; Xing et al., 2024). In line with this, a magnitude-based pruning of 30% of visual tokens yields a similar 6% speedup gain. Third, adaptive target feature injection explores the impact of feature extraction positions within the target model. While prior speculative decoding methods such as EAGLE (Li et al., 2024c;d) extract features from fixed layers; our approach systematically searches for optimal extraction strategies tailored to each configuration.

In addition, we explore the choice of layer in the target model $M_{ta}$ from which features are extracted for the cross-attention mechanism in the draft model $M_{da}$, as illustrated in the *Adaptive Target Feature Injection* module in Figure 2(b). The OFA search process is illustrated in Figure 2(c), where during each training iteration of STAR, a draft subnetwork with a specific visual token budget $r*$, attention head configuration $\mathbf{H} = \{H_j\}, 1 \le j \le M$, and selected connection layer $l$ is randomly sampled. This draft model, denoted as $M_{da}(r, \mathbf{H}, l)$, is then trained following the procedure described in Figure 2(b). Next, we detail the search dimensions involved in this process.

**Attention Head-Wise Pruning.** STAR dynamically computes head importance without storing persistent rankings. Let $H'_j$ (where $1 \le H'_j \le H_j$) represent the number of attention heads retained at the $j$-th layer. During training, for each subnetwork configuration, we select and retain a specific top $H'_j$ attention heads with the highest importance scores. To evaluate the importance $I_j$ of each attention head $h$, we aggregate the product of gradients and their corresponding weight magnitudes across all projection matrices associated with that head (Michel et al., 2019; Molchanov et al., 2016):

$$I_j = \sum_{P \in \{Q,K,V\}} \sum_{x,y} \left| \nabla W_P^{h,j}[x,y] \cdot W_P^{h,j}[x,y] \right| \tag{1}$$

Here, $W_P^{h,j}$ denotes a projection matrices (query (Q), key (K), or value (V)) associated with head $h$ at layer $j$, and $\nabla W_P^{h,j}$ represents its corresponding gradient. The notation $[x,y]$ refers to the element located at the $x$-th row and $y$-th column of the matrix. In each training iteration, attention heads are ranked using Equation 1, and a subset is selected according to the specified budget $H_j$.

**Visual Token Compression.** STAR adopts a target-aware token selection mechanism that leverages the attention patterns of $M_{ta}$ to guide this visual token selection. STAR evaluates the visual token importance using attention scores from the target model's final layer during the prefilling phase: Given target model attention weights $A^{(L)} \in \mathbb{R}^{B \times H \times Q \times K}$ from the final layer, the importance score for visual token $j$ is:

$$I_j = \frac{1}{H \cdot Q} \sum_{h=1}^{H} \sum_{i=1}^{Q} A_{h,i,j}^{(L)} \tag{2}$$

In each forward pass, a token budget $r$ is randomly sampled from the budget pool $\mathcal{R}$. Based on this budget, only $r$ visual tokens are retained from the input according to their importance scores, while the remaining tokens remain unchanged. The draft model is trained to produce the same output as the target model, even with fewer visual tokens. This selective compression strategy preserves the most informative visual features while lowering computational cost.

**Adaptive Target Feature Injection.** STAR explore the performance impact of selecting different target layers. We consistently use final-layer target features for supervision during warmup training, then systematically explore different target layer choices during multi-resolution training to optimize draft performance within our NAS framework (Figure 2(b)). In each training iteration, a target feature is randomly sampled from one of the candidate layers in the target model and injected into the draft model. To incorporate this feature, the draft model includes a cross-attention layer inserted at a predefined layer index. Specifically, we integrate the target features from a selected layer $l'$ of $M_{ta}$, denoted as $S^{l'} = (s_1^{l'}, s_2^{l'}, \ldots, s_n^{l'})$, with the intermediate features of the $j$-th layer of the draft model $M_{da}$, denoted as $E^j = (e_1^j, e_2^j, \ldots, e_{n'}^j)$, via a cross-attention mechanism. In this formulation, $E^j$ serves as the query, while $S^{l'}$ acts as the keys and values.

### 3.2 Adaptive Intermediate Feature Distillation

Beyond the NAS framework detailed in Section 3.1, STAR adaptively selects intermediate features from the target model $M_{ta}$ for distillation into the draft model's early layers using a dedicated loss function. To effectively guide the training of the draft model $M_{da}$, the selected target features must meet two essential criteria. They should capture semantically meaningful content and exhibit low variability across layers to ensure stable learning. Prior studies (Sun et al., 2020; Skean et al., 2025; Jain et al., 2024) show that intermediate features with low attention entropy and high consistency provide stable supervision signals. STAR adopts a simple yet effective strategy to identify such features from each layer of the target model, as illustrated in Figure 2(b), to support the efficient training of the draft model. However, low entropy alone is insufficient since a layer may exhibit low entropy while fluctuating strongly between adjacent layers, causing unstable supervision. STAR therefore jointly considers both the entropy value ($\text{AE}(\ell)$) and its inter-layer variation $\Delta\text{AE}(\ell)$, selecting layers that are information rich and consistent across depth for robust knowledge transfer.

Specifically, for the $l$-th decoder block of $M_{ta}$, its input tokens and output tokens are denoted as $S^{\ell-1} = (s_1^{\ell-1}, s_2^{\ell-1}, \ldots, s_n^{\ell-1})$ and $S^\ell = (s_1^\ell, s_2^\ell, \ldots, s_n^\ell)$, respectively. Let the attention matrix $A_\ell$ associated with $l$-th layer as $A_\ell = \text{softmax}(\frac{Q_\ell K_\ell^\top}{\sqrt{z}})$, where $Q_\ell = S^{\ell-1} W_Q$ and $K_\ell = S^{\ell-1} W_K$, and $z$ denotes the hidden dimension of the $M_{da}$ The average attention entropy (AE) is calculated as $\text{AE}(\ell) = -\frac{1}{n} \sum_{i=1}^n \sum_{j=1}^n A_{\ell,i,j} \log A_{\ell,i,j}$ where $A_{\ell,i,j}$ denotes the $(i,j)$-th element of $A_\ell$. In practice with multiple heads, $\text{AE}(\ell)$ is also averaged across the selected attention heads of the subnetwork. To capture variation across layers, we further define the one-step difference $\Delta\text{AE}(\ell) = |\text{AE}(\ell) - \text{AE}(\ell-1)|$. By jointly considering both $\text{AE}(\ell)$ and its inter-layer variation $\Delta\text{AE}(\ell)$, we identify the optimal distillation layer $\ell_d^* = \arg\min_{\ell \in L} [\Delta\text{AE}(\ell) + \text{AE}(\ell)]$, ensuring transferred features from $M_{ta}$ to $M_{da}$ are semantically rich and locally stable.

### 3.3 TPPT Loss Design

This section presents the design of the TPPT loss function with a summary of the training algorithm. We employ a multi-component weighted loss function to align the draft model with the target model across multiple levels of representation, where $\lambda$ terms control the relative importance of each component. The loss function comprises three terms: (1) a KL divergence loss $\mathcal{L}_{\text{KL}} = \text{KL}(\text{softmax}(D), \text{softmax}(T))$ that ensures output token distributions match between the draft and target models, where $D = (d_1, \ldots, d_n)$ and $T = (t_1, \ldots, t_n)$ represent the predicted token logits from the draft model $M_{da}$ and target model $M_{ta}$, respectively; (2) an intermediate feature distillation loss $\mathcal{L}_{distill} = \text{smoothL1}(E^m, S^{\ell^\star})$ that aligns early-layer features from the draft model ($E^m$ with $m = 1$) with adaptively selected intermediate features from the target model ($S^{\ell_d^*}$); and (3) a feature alignment loss $\mathcal{L}_{feat} = \text{smoothL1}(E^M, S^L)$ that matches the final-layer features between the draft model's output $E^M = (e_1^M, \ldots, e_n'^M)$ and the target model's output $S^L = (s_1^L, \ldots, s_n^L)$ to improve token acceptance rates. The overall loss $\mathcal{L}_{final}$ for the TPPT is:

$$\mathcal{L}_{final} = \lambda_{feat} \mathcal{L}_{feat} + \lambda_{intermed} \mathcal{L}_{intermed} + \lambda_{KL} \mathcal{L}_{\text{KL}} \tag{3}$$

## 4 Results

**Experimental Setup.** We assess STAR across four VLMs spanning different parameter scales: LLaVA-v1.6-Vicuna (7B, 13B) (Liu et al., 2024b), Pixtral (12B) (Agrawal et al., 2024) and SmolVLM (2B) (Marafioti et al., 2025). Evaluation is conducted on six multimodal benchmarks: MMT-Bench (Ying et al., 2024), SEED-Bench-2 (Li et al., 2023), ScienceQA (Lu et al., 2022), OCRBench (Liu et al., 2024c), ChartQA (Masry et al., 2022), and MathVista (Lu et al., 2024). We measure two primary metrics: (1) **Speedup ratio** calculated as $t_{\text{AR}}/t_{\text{method}}$, where $t_{\text{AR}}$ represents the average wall-clock time per token for standard autoregressive decoding and $t_{\text{method}}$ denotes the time for each evaluated approach. Higher speedup values indicate reduced end-to-end latency. (2) **Average token acceptance length** $\tau$, quantifying consecutive draft tokens accepted during verification. Larger $\tau$ values indicate fewer verification rounds and improved throughput. We implement six state-of-the-art SD methods adapted for VLMs: SPD (Gagrani et al., 2024), Kangaroo (Liu et al., 2024a), Medusa (Cai et al., 2024), Hydra (Ankner et al., 2024), and EAGLE 1 and 2 (Li

Table 2: Evaluation of STAR on speedup ratio (S) and average accepted token length ($\tau$).

| Models | Methods | MMT | | SEED | | ScienceQA | | OCRBench | | ChartQA | | MathVista | | Average | |
|---|---|---|---|---|---|---|---|---|---|---|---|---|---|---|---|
| | | S | $\tau$ | S | $\tau$ | S | $\tau$ | S | $\tau$ | S | $\tau$ | S | $\tau$ | S | $\tau$ |
| LLaVA-v1.6 Vicuna-7B | SPD (Gagrani et al., 2024) | 1.10 | 1.88 | 0.81 | 1.17 | 1.08 | 1.87 | 0.89 | 1.25 | 0.91 | 1.24 | 1.06 | 1.76 | 0.97 | 1.53 |
| | Kangaroo (Liu et al., 2024a) | 1.32 | 2.11 | 1.33 | 2.12 | 1.31 | 2.09 | 1.17 | 1.89 | 1.18 | 1.98 | 1.15 | 1.86 | 1.24 | 2.01 |
| | Medusa (Cai et al., 2024) | 1.58 | 2.88 | 1.59 | 3.01 | 1.44 | 2.77 | 1.22 | 2.33 | 1.25 | 2.41 | 1.22 | 2.34 | 1.38 | 2.62 |
| | Hydra (Ankner et al., 2024) | 1.78 | 3.86 | 1.72 | 3.88 | 1.68 | 3.79 | 1.41 | 3.21 | 1.35 | 3.11 | 1.42 | 3.25 | 1.56 | 3.52 |
| | EAGLE (Li et al., 2024c) | 2.10 | 5.04 | 2.09 | 5.01 | 1.98 | 4.88 | 1.72 | 4.13 | 1.56 | 3.98 | 1.78 | 4.25 | 1.87 | 4.55 |
| | EAGLE-2 (Li et al., 2024d) | 2.31 | 5.48 | 2.31 | 5.61 | 2.15 | 5.22 | 1.92 | 4.88 | 1.77 | 4.22 | 1.87 | 4.67 | 2.05 | 5.01 |
| | EAGLE-3 (Li et al., 2025) | 2.38 | 5.72 | 2.36 | 5.82 | 2.22 | 5.52 | 2.02 | 5.24 | 1.83 | 4.46 | 1.97 | 5.02 | 2.13 | 5.30 |
| | **STAR** | **2.67** | **6.27** | **2.61** | **6.18** | **2.45** | **5.71** | **2.11** | **4.89** | **2.04** | **4.39** | **2.20** | **5.30** | **2.35** | **5.46** |
| LLaVA-v1.6 Vicuna-13B | SPD | 1.07 | 1.78 | 1.06 | 1.79 | 1.09 | 1.88 | 0.86 | 1.12 | 0.89 | 1.25 | 0.87 | 1.22 | 1.00 | 1.58 |
| | Kangaroo | 1.43 | 1.77 | 1.51 | 1.87 | 1.22 | 1.55 | 1.21 | 1.54 | 1.27 | 1.61 | 1.53 | 2.01 | 1.36 | 1.72 |
| | Medusa | 1.99 | 2.67 | 1.96 | 2.76 | 1.93 | 2.77 | 1.40 | 2.92 | 1.51 | 2.82 | 1.51 | 2.62 | 1.72 | 2.76 |
| | Hydra | 2.12 | 2.87 | 2.08 | 2.99 | 2.21 | 3.12 | 1.49 | 3.07 | 1.65 | 3.03 | 1.66 | 2.87 | 1.87 | 2.99 |
| | EAGLE | 2.45 | 3.56 | 2.19 | 3.24 | 2.63 | 3.98 | 1.65 | 3.31 | 1.85 | 3.27 | 1.8 | 3.09 | 2.10 | 3.41 |
| | EAGLE-2 | 2.89 | 4.05 | 3.18 | 4.33 | 3.09 | 4.97 | 2.20 | 4.12 | 2.41 | 4.15 | 2.39 | 3.76 | 2.69 | 4.23 |
| | EAGLE-3 | 3.45 | 4.90 | 3.34 | 4.65 | 3.19 | 5.20 | 2.50 | 4.79 | 2.46 | 4.37 | 2.42 | 3.85 | 2.89 | 4.63 |
| | **STAR** | **3.85** | **5.56** | **3.61** | **5.32** | **3.41** | **5.19** | **2.77** | **4.61** | **2.67** | **4.17** | **2.62** | **4.11** | **3.16** | **4.82** |
| Pixtral-12B | SPD | 1.08 | 1.51 | 1.03 | 1.47 | 1.05 | 1.49 | 1.05 | 1.49 | 1.04 | 1.43 | 1.04 | 1.46 | 1.05 | 1.47 |
| | Kangaroo | 1.26 | 1.54 | 1.09 | 1.39 | 1.14 | 1.51 | 1.16 | 1.52 | 1.12 | 1.47 | 1.13 | 1.49 | 1.15 | 1.49 |
| | Medusa | 1.37 | 1.81 | 1.37 | 1.81 | 1.35 | 1.87 | 1.24 | 1.69 | 1.22 | 1.68 | 1.16 | 1.47 | 1.28 | 1.72 |
| | Hydra | 1.58 | 2.24 | 1.47 | 2.04 | 1.53 | 2.06 | 1.38 | 1.81 | 1.34 | 1.79 | 1.36 | 1.78 | 1.44 | 1.95 |
| | EAGLE | 2.38 | 3.47 | 1.97 | 2.53 | 2.31 | 3.64 | 1.69 | 2.73 | 1.78 | 2.84 | 1.64 | 2.47 | 1.96 | 2.95 |
| | EAGLE-2 | 2.81 | 3.95 | 2.31 | 3.07 | 2.64 | 4.03 | 2.12 | 3.25 | 2.14 | 3.17 | 1.81 | 2.73 | 2.31 | 3.37 |
| | EAGLE-3 | 2.83 | 4.12 | 2.46 | 3.40 | 2.79 | 4.41 | 2.22 | 3.48 | 2.26 | 3.51 | 2.13 | 3.38 | 2.45 | 3.72 |
| | **STAR** | **3.01** | **4.41** | **2.73** | **3.56** | **3.09** | **3.93** | **2.46** | **3.44** | **2.40** | **3.42** | **2.42** | **3.34** | **2.69** | **3.68** |
| SmolVLM-2B | SPD | 1.02 | 1.33 | 1.04 | 1.41 | 1.06 | 1.43 | 1.06 | 1.42 | 1.07 | 1.46 | 1.02 | 1.34 | 1.04 | 1.40 |
| | Kangaroo | 1.28 | 1.48 | 1.08 | 1.18 | 1.03 | 1.17 | 1.06 | 1.22 | 1.04 | 1.14 | 1.08 | 1.23 | 1.10 | 1.24 |
| | Medusa | 2.12 | 2.71 | 1.51 | 2.00 | 1.72 | 2.22 | 1.20 | 1.61 | 1.15 | 1.55 | 1.35 | 1.75 | 1.51 | 1.97 |
| | Hydra | 2.33 | 3.07 | 1.62 | 2.08 | 1.98 | 2.66 | 1.32 | 1.74 | 1.22 | 1.58 | 1.51 | 1.98 | 1.66 | 2.19 |
| | EAGLE | 2.57 | 3.42 | 1.85 | 2.56 | 2.16 | 2.76 | 1.42 | 1.88 | 1.34 | 1.77 | 1.65 | 2.22 | 1.83 | 2.44 |
| | EAGLE-2 | 2.96 | 3.89 | 2.12 | 2.93 | 2.39 | 3.21 | 1.65 | 2.11 | 1.51 | 2.13 | 1.81 | 2.63 | 2.07 | 2.82 |
| | EAGLE-3 | 3.00 | 3.94 | 2.17 | 3.04 | 2.65 | 3.57 | 1.78 | 2.33 | 1.60 | 2.30 | 1.97 | 2.84 | 2.20 | 3.00 |
| | **STAR** | **3.12** | **3.94** | **2.28** | **3.16** | **2.91** | **3.57** | **1.88** | **2.51** | **1.64** | **2.28** | **2.06** | **2.82** | **2.32** | **3.05** |
| | Temperature = 1 | | | | | | | | | | | | | | |
| LLaVA-v1.6 Vicuna-7B | SPD | 0.83 | 1.19 | 0.81 | 1.15 | 0.85 | 1.18 | 0.75 | 1.06 | 0.72 | 1.08 | 0.92 | 1.48 | 0.81 | 1.19 |
| | Kangaroo | 1.20 | 1.97 | 1.26 | 2.03 | 1.23 | 2.01 | 1.09 | 1.80 | 1.11 | 1.89 | 1.07 | 1.77 | 1.16 | 1.91 |
| | EAGLE-2 | 2.19 | 5.37 | 2.20 | 5.48 | 2.04 | 5.12 | 1.82 | 4.77 | 1.68 | 4.13 | 1.76 | 4.56 | 1.95 | 4.91 |
| | EAGLE-3 | 2.25 | 5.70 | 2.25 | 5.72 | 2.10 | 5.38 | 1.89 | 5.01 | 1.71 | 4.28 | 1.88 | 4.98 | 2.01 | 5.18 |
| | **Star** | **2.50** | **6.25** | **2.45** | **5.92** | **2.33** | **5.56** | **2.03** | **4.75** | **1.97** | **4.22** | **2.09** | **5.13** | **2.23** | **5.30** |
| LLAVA-v1.6 Vicuna-13B | SPD | 0.88 | 1.22 | 0.84 | 1.25 | 0.84 | 1.32 | 0.79 | 1.18 | 0.81 | 1.14 | 0.88 | 1.24 | 0.84 | 1.22 |
| | Kangaroo | 1.23 | 1.57 | 1.17 | 1.53 | 1.07 | 1.44 | 1.01 | 1.24 | 1.07 | 1.34 | 1.21 | 1.67 | 1.13 | 1.46 |
| | EAGLE-2 | 2.35 | 3.75 | 3.02 | 4.30 | 3.03 | 4.67 | 2.03 | 3.87 | 2.18 | 3.83 | 2.18 | 3.41 | 2.46 | 3.97 |
| | EAGLE-3 | 2.92 | 4.77 | 3.12 | 4.61 | 3.06 | 4.89 | 2.08 | 4.03 | 2.26 | 4.04 | 2.19 | 3.55 | 2.61 | 4.32 |
| | **STAR** | **3.51** | **5.37** | **3.55** | **5.00** | **3.38** | **5.88** | **2.35** | **3.92** | **2.59** | **4.09** | **2.38** | **3.99** | **2.96** | **4.71** |
| Pixtral-12B | SPD | 0.81 | 1.15 | 0.79 | 1.11 | 0.80 | 1.12 | 0.80 | 1.13 | 0.75 | 1.07 | 0.77 | 1.09 | 0.79 | 1.11 |
| | Kangaroo | 1.18 | 1.41 | 1.08 | 1.35 | 1.03 | 1.36 | 1.19 | 1.48 | 1.14 | 1.45 | 1.09 | 1.41 | 1.12 | 1.41 |
| | EAGLE-2 | 2.76 | 3.81 | 2.24 | 3.01 | 2.76 | 3.87 | 2.23 | 3.24 | 2.03 | 3.09 | 1.79 | 2.69 | 2.30 | 3.28 |
| | EAGLE-3 | 2.79 | 4.02 | 2.33 | 3.25 | 2.80 | 4.03 | 2.25 | 3.51 | 2.27 | 3.58 | 1.92 | 2.98 | 2.39 | 3.56 |
| | **STAR** | **2.98** | **3.93** | **2.56** | **3.48** | **2.99** | **3.79** | **2.34** | **3.32** | **2.26** | **3.09** | **2.22** | **3.22** | **2.56** | **3.47** |
| SmolVLM-2B | SPD | 1.07 | 1.47 | 1.01 | 1.33 | 1.07 | 1.46 | 0.97 | 1.26 | 1.06 | 1.44 | 0.85 | 1.20 | 1.00 | 1.36 |
| | Kangaroo | 1.37 | 1.59 | 1.12 | 1.24 | 1.22 | 1.41 | 1.12 | 1.29 | 1.18 | 1.36 | 1.28 | 1.42 | 1.22 | 1.39 |
| | EAGLE-2 | 2.62 | 3.60 | 1.92 | 2.67 | 2.24 | 3.11 | 1.41 | 1.77 | 1.60 | 2.18 | 1.77 | 2.49 | 1.93 | 2.64 |
| | EAGLE-3 | 2.77 | 3.82 | 2.11 | 3.04 | 2.63 | 3.65 | 1.46 | 1.90 | 1.64 | 2.29 | 1.84 | 2.64 | 2.08 | 2.89 |
| | **STAR** | **2.93** | **3.61** | **2.33** | **3.30** | **2.96** | **3.67** | **1.59** | **2.12** | **1.81** | **2.48** | **2.01** | **2.66** | **2.27** | **2.97** |

et al., 2024c;d). Target VLMs remain frozen while only draft models undergo training. We utilize the LLaVA-`mix665k` dataset with 55,000 training samples, supplemented by 1,000 samples from each evaluation benchmark that are **disjoint** from the test sets for domain adaptation. For both

phases within TPPT, the training proceeds for 68,000 iterations using AdamW optimizer ($\beta_1 = 0.9$, $\beta_2 = 0.95$ with a learning rate of $3 \times 10^{-5}$ and gradient clipping at 0.5.

For multi-resolution training in TPPT, the visual token pruning budget pool is defined as $\mathcal{R} = \{0.1n, 0.2n, \ldots, n\}$, where $n$ denotes the total number of prompt tokens. The head pruning configuration is set as $H_j = \{0.25H, 0.5H, 0.75H, H\}$, where $H_j$ is the number of retained attention heads and $H$ is the total number of heads. Adaptive token feature injection searches across the last five layers of the target model to determine the optimal layer index for injecting features into the draft model. After TPPT is complete, the optimal draft model is selected by exhaustively searching through all subnetwork candidates to identify the one that achieves the highest speedup. The draft architecture comprises three decoder layers. Loss term weights are configured as: $\mathcal{L}_{\text{feat}} = 0.2$, $\mathcal{L}_{\text{distill}} = 0.2$, and $\mathcal{L}_{\text{KL}} = 1.0$. All experiments run on a single NVIDIA A100 80GB GPU.

For the main results in Table 2, we measure decoding speed with an inference batch size of 1, following standard practice in speculative decoding, and report wall-clock time per generated token over the full decoding pipeline. All methods (STAR and SPD, Kangaroo, Medusa, Hydra, EAGLE-1, EAGLE-2) are implemented and evaluated in PyTorch 2.0.1 with HuggingFace Transformers 4.36.2 under CUDA 12.4, without additional inference frameworks such as vLLM or TensorRT-LLM. The hardware comparison in Table 3 further evaluates the same implementations on NVIDIA H100 80GB and RTX8000 48GB GPUs under the same software configuration.

The TPPT framework involves a comprehensive search process to identify optimal architectures across diverse configurations. However, this one-time training overhead remains small. Using LLaVA-v1.6-Vicuna-7B on four NVIDIA A100 80GB GPUs, Phase 1 supernet training requires approximately 3.5 hours per epoch, and Phase 2 subnet training takes around 4.5 hours due to additional dynamic pruning operations. The subsequent exhaustive search to identify optimal subnetworks varies by dataset complexity. For instance, evaluating each NAS configuration on MMT-Bench requires approximately 12 minutes per 100 mini-batches on a single NVIDIA A100 GPU. This search is conducted offline once per model–hardware pair to select the final draft configuration, and is not repeated during deployment. As a result, the search cost is fully amortized over all subsequent inference on that model–hardware pair and remains negligible compared to TPPT training and standard VLM pretraining or fine-tuning.

### 4.1 MAIN RESULTS AND DISCUSSION

Table 2 demonstrates STAR's performance across four VLMs and six multimodal benchmarks, showing consistent superiority over existing speculative decoding methods in both speedup ratios S and token acceptance lengths $\tau$. STAR achieves substantial acceleration across all evaluated models, with average speedups ranging from $2.32\times$ to $3.16\times$ compared to standard autoregressive decoding. Most notably, STAR outperforms the strongest baseline EAGLE-3 by significant margins: 10% LLaVA-7B ($2.35\times$ vs $2.13\times$), 9% on LLaVA-13B ($3.16\times$ vs $2.89\times$), 10% on Pixtral-12B ($2.69\times$ vs $2.45\times$), and 5% on SmolVLM-2B ($2.32\times$ vs $2.20\times$). The token acceptance lengths follow similar trends, with STAR achieving $\tau$ values of 5.46, 4.82, 3.68, and 3.05 respectively. Interestingly, while STAR's token acceptance lengths are sometimes comparable to or slightly lower than EAGLE-3 (e.g., 5.46 vs 5.30 on LLaVA-7B, 4.82 vs 4.63 on LLaVA-13B), STAR achieves higher speedups through attention head pruning and visual token compression, which reduce draft model FLOPs. This creates a favorable trade-off where slightly lower or comparable draft quality is more than compensated by substantially more efficient token generation. For instance, on LLaVA-13B, STAR achieves $3.16\times$ speedup with $\tau = 4.82$, while EAGLE-3 achieves $2.89\times$ speedup despite comparable $\tau = 4.63$, demonstrating the effectiveness of STAR's architectural optimizations.

Larger models demonstrate greater benefits from STAR's optimizations. The 13B LLaVA model achieves the highest speedup of $3.16\times$, compared to $2.35\times$ for the 7B variant. This aligns with the expectation that computationally heavier target models create more opportunities for draft model acceleration. Pixtral-12B shows competitive performance despite its larger parameter count, suggesting that STAR's target-aware compression effectively handles diverse architectural designs.

STAR exhibits varying effectiveness across different benchmark types. Vision-language reasoning tasks like MMT-Bench and ScienceQA consistently yield the highest speedups (e.g., $2.67\times$ and $2.45\times$ respectively on LLaVA-7B), as these tasks benefit from STAR's ability to capture semantic relationships between visual and textual content. Conversely, tasks requiring precise visual detail

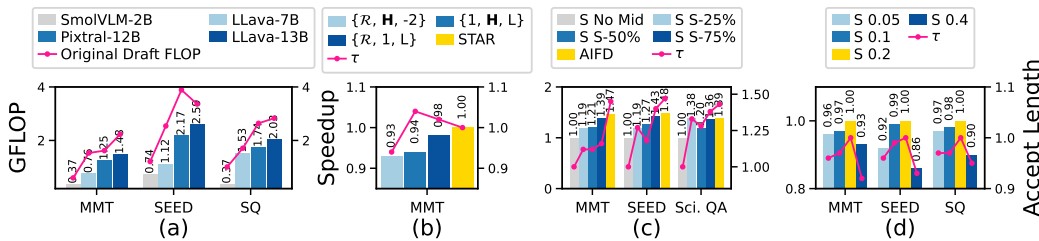

Figure 3: (a) FLOPs of the selected draft models. (b) STAR performance in various NAS settings. (c) Evaluation of AIFD. (d) Impact of $\lambda$ on STAR performance. In (b)–(d), bars show speedup (left axis) and red curve shows acceptance length (right axis).

recognition, such as OCRBench, show more modest improvements ($2.11\times$ on LLaVA-7B), reflecting the inherent difficulty of compressing visual information without accuracy loss.

SPD and Kangaroo show limited effectiveness for multimodal tasks, often achieving speedups barely above $2.08\times$. Multi-head approaches (Medusa, Hydra) perform better but remain substantially below STAR's performance. The EAGLE series, is the strongest competition but lacks STAR's multimodal-specific optimizations, resulting in lower performance across all evaluated scenarios.

Figure 3(a) illustrates the diversity of optimal draft models discovered by STAR across different datasets for $T = 0$. Specifically, we plot the FLOPs of the searched optimal draft models alongside those of the original draft models without attention head pruning or visual token pruning. For example, on LLaVA-7B, the optimal draft configuration requires only 0.76 GFLOPs on MMT-Bench versus 1.53 GFLOPs for the full model. This comparison highlights how STAR adapts the draft model architecture to different data distributions for improved speedup highlighted in Table 2.

Temperature settings critically impact performance, with STAR achieving optimal results at deterministic decoding $T = 0$ but degrading moderately under stochastic sampling $T = 1$ due to increased token variance. Nevertheless, on LLaVA-v1.6-Vicuna-7B, STAR maintains superior performance with $2.23\times$ speedup and $\tau = 5.30$ versus EAGLE-3's $2.01\times$ speedup and $\tau = 5.18$.

## 4.2 Ablation Studies

**Impact of NAS searching dimension.** To evaluate the impact of each NAS search dimension, we conduct ablation studies on LLaVA-v1.6-Vicuna-7B using MMT-Bench. Specifically, we assess the contributions of searching visual token pruning ($r$), attention head pruning ($H'_j$), and adaptive target feature injection ($l'$) by disabling each dimension individually. The results are shown in Figure 3(b), where $\{\mathcal{R}, \mathbf{H}, -2\}$ denotes searching only for the optimal visual token and head pruning while fixing the draft injection to the second-to-top layer. Similarly, $\{1, \mathbf{H}, L\}$ and $\{\mathcal{R}, 1, L\}$ indicate disabling visual token and head pruning search. Figure 3(b) shows that STAR achieves the highest speedup ($2.67\times$) while maintaining competitive token acceptance ($\tau = 6.27$). Removing head pruning dimension reduces speedup to $2.62\times$ but increases acceptance length to $\tau = 6.44$, as the larger draft model with full attention heads captures more information but incurs higher computational cost. Removing visual pruning dimension further decreases speedup to $2.51\times$ while achieving the highest acceptance length ($\tau = 6.50$), since processing all visual tokens provides complete visual context at the expense of increased latency. Using the target feature injection at fixed position degrades performance ($2.48\times$ speedup), showing that adaptive target feature injection is crucial.

**Evaluation of AIFD.** We evaluate the impact of the adaptive intermediate feature distillation (AIFD) strategy described in Section 3.2 on STAR performance. Experiments are conducted using the LLaVA-v1.6-Vicuna-7B model across MMT-Bench, SEED-Bench, and ScienceQA datasets. To show its

Table 3: GPU performance comparison: Eagle vs STAR.

| GPUs | Eagle2 | | STAR | |
|---|---|---|---|---|
| | Speedup | Tokens/S | Speedup | Tokens/S |
| A100 | $2.26\times$ | 82.48 | $2.58\times$ | 94.43 |
| H100 | $2.60\times$ | 138.52 | $2.99\times$ | 153.12 |
| RTX8000 | $1.83\times$ | 36.43 | $2.23\times$ | 43.73 |

advantage, we design four baselines:

(1) *No Mid Tuning* (No Mid), which trains STAR without intermediate features; (2) *Static-25%* (S-25%), using features from fixed position of target model's 25% depth; (3) *Static-50%* (S-50%) extracting from 50% depth; and (4) *Static-75%* (S-75%) utilizing 75% depth features.

Figure 3(c) presents the results. Training without intermediate supervision yields the lowest performance across all metrics. Static-25% and Static-50% show comparable performance, while deeper layers provide progressively better guidance, with Static-75% achieving the strongest static results. However, AIFD consistently outperforms all static baselines, confirming that adaptive layer selection based on attention entropy and stability effectively identifies the most informative supervision signals for draft model training.

**Adaptivity of STAR across Different Hardware Conditions.** Unlike existing SD baselines that are agnostic to hardware conditions, STAR adapts to different hardware platforms by searching the subnetwork space and adjusting draft model configurations, achieving superior performance across diverse hardware conditions. To show this, we evaluate STAR across three different GPU architectures: Nvidia RTX8000 48GB, Nvidia A100 80GB, and Nvidia H100 80GB, representing different hardware conditions. The experiments are conducted on the LLaVA-v1.6-Vicuna-7B model across the MMT-Bench, SEED-Bench, and ScienceQA datasets. Table 3 indicates that STAR consistently outperforms EAGLE-2 across all hardware configurations. On high-performance GPUs like H100, STAR achieves $2.99\times$ speedup compared to EAGLE-2's $2.60\times$, with throughput reaching 153.12 tokens/s versus 138.52 tokens/s. On resource-constrained hardware like RTX8000, STAR achieves $2.23\times$ speedup while EAGLE-2 drops to $1.83\times$, showing STAR's robustness across diverse hardware environments.

**Impact of Lambda Setting.** As described in equation 3, the loss weights $\lambda_{feat}$, $\lambda_{distill}$, and $\lambda_{KL}$ control the relative importance of the loss function. Since $\mathcal{L}_{feat}$ and $\mathcal{L}_{distill}$ are both smooth L1 losses operating at similar scales with comparable roles in feature alignment, we set $\lambda_{feat} = \lambda_{distill}$ to simplify hyperparameter tuning. The KL divergence loss maintains consistent influence with $\lambda_{KL} = 1$. Figure 3(d) demonstrates the impact of varying $\lambda_{feat}$ on LLaVA-v1.6-Vicuna-7B performance across three benchmarks. Each number represents the static (S) value for both $\lambda_{feat}$ and $\lambda_{distill}$, while $\lambda_{KL}$ is fixed to 1. Increasing $\lambda_{feat}$ from 0.05 to 0.2 consistently improves performance metrics across all datasets, indicating that stronger feature supervision enhances draft model quality. However, further increasing to 0.4 leads to performance degradation, suggesting that excessive feature supervision can impair the model's ability to generalize effectively. This validates our choice of $\lambda_{feat} = \lambda_{distill} = 0.2$, as the optimal balance point.

## 5 CONCLUSION

In this paper, we introduce *STAR*, a speculative decoding framework optimized for vision-language models. By combining neural architecture search and attention-guided feature distillation, STAR achieves up to $3.8\times$ speedup over the existing SD baseline while preserving task performance across diverse multimodal benchmarks. Our results highlight the effectiveness of STAR for fast, scalable multimodal inference and demonstrate that hardware-aware, target-guided draft design is a practical and robust path toward accelerating future VLM deployments.

## 6 LIMITATIONS

Our contribution is primarily system-level and application-driven: rather than proposing a new generic NAS algorithm or theoretical decoding principle, we instantiate a multimodal, hardware-aware search space for draft design and, given its current moderate size, adopt exhaustive evaluation; as the space scales to richer architectures or devices, STAR can naturally incorporate more advanced strategies such as OFA-style predictor-based search, remaining orthogonal and complementary to future advances in speculative decoding methods.

## ETHICS STATEMENT

All authors adhere to the ICLR Code of Ethics. This work presents a computational optimization technique for vision-language models using publicly available datasets and models. The proposed STAR framework improves computational efficiency without altering model capabilities or introducing new risks. The research focuses on technical acceleration methods and does not raise concerns regarding bias, fairness, privacy violations, or potential harmful applications.

## REPRODUCIBILITY STATEMENT

Complete source code for STAR implementation is provided in supplementary materials. Section 4 provides comprehensive experimental details including hyperparameters, training procedures, and hardware specifications. All experiments use publicly available datasets with specified preprocessing steps. Baseline methods are implemented following their original papers for comparison with STAR. Results are reported across multiple runs to ensure reliability.

## 7 USE OF LARGE LANGUAGE MODELS

LLMs were used solely for grammar checking and minor language improvements. Usage was limited to proofreading and correcting grammatical errors. LLMs were not involved in research ideation, methodology design, experimental analysis, or results interpretation. All content remains the work of human authors who take full responsibility for the manuscript.

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

## A  APPENDIX

The complete algorithm for our Two-Phase Progressive Training (TPPT) framework is presented in Algorithm 1. This algorithm encompasses both the warm-up training phase (Phase 1) and the multi-resolution training phase (Phase 2) described in Section 3.1.

To validate the sufficiency of our chosen search space granularity, we conduct experiments comparing standard and doubled search resolution configurations on LLaVA-v1.6-Vicuna-7B across three benchmarks (Table 4). During the TPPT training phase, both standard and doubled granularity configurations utilize their respective complete search spaces. Our standard Head Pruning Ratio trains with search space $\{0.25, 0.5, 0.75, 1.0\}$, while the Double Head Pruning Ratio configuration trains with the finer grained space $\{0.125, 0.25, 0.375, 0.5, 0.625, 0.75, 0.875, 1.0\}$. Similarly, our standard Vision Compression Ratio trains with the budget pool $\mathcal{R} = \{0.1n, 0.2n, \ldots, n\}$, whereas the Double Vision Compression Ratio trains with $\{0.05n, 0.1n, 0.15n, \ldots, n\}$. After training, we perform exhaustive NAS search across all trained subnetworks to identify the optimal configuration for each benchmark. https://docs.google.com/document/d/1Lcvzm48zDczDiioMK4FWWIuho-p84M1zKBL7DtyThe0/edit?tab=t.pk6znxnmmek7, where MMT Bench uses 8,000 training samples, SEED Bench uses 8,000 samples, ScienceQA uses 6,000 samples, OCRBench uses 8,000 samples, ChartQA uses 8,000 samples and MathVista uses 5,000 samples, with all benchmarks evaluated on 1,000 test samples.

---

**Algorithm 1** Two-Phase Progressive Training for drafting (TPPT)

---

**Require:** Training dataset $\mathcal{D}$, supernet draft model $\mathcal{M}_{da}$, target model $\mathcal{M}_{ta}$
**Output:** Trained model with optimal subnet architectures
  Initialize supernet parameters $\theta$
  **Phase 1: Supernet Training**
  **for** each supernet training epoch **do**
    **for** each batch $\mathcal{B} \in \mathcal{D}$ **do**
      $T, S \leftarrow \mathcal{M}_{ta}(\mathcal{B})$ {Target model forward pass}
      $S^L \leftarrow S$ {Target's final hidden features}
      $S^{\ell^\star} \leftarrow \text{AIFD}(S)$ {Adaptive Intermediate Layer distillation}
      $D, E^m, E^M \leftarrow \mathcal{M}_{da}(\mathcal{B}, S^L)$ {Draft model forward pass}
      Compute Loss$(S^{\ell^\star}, E^m, S^L, E^M, T, D)$
      Update $\theta$
  **Phase 2: Subnet Training**
  **for** each subnet training epoch **do**
    **for** each batch $\mathcal{B} \in \mathcal{D}$ **do**
      $T, S \leftarrow \mathcal{M}_{ta}(\mathcal{B})$
      $S^{\ell^\star} \leftarrow AIFD(S)$
      Sample $r, H'_j, l'$ From $\mathcal{R}, \mathbf{H}, L$
      $\mathcal{B}' \leftarrow \text{Prune}(\mathcal{B}, r)$ {Apply Visual Token Compression}
      $S^{l'} \leftarrow S$ {Extract Target's feature from selected layer}
      $D, E^m, E^M \leftarrow \mathcal{M}_{da}(\mathcal{B}', H'_j, S^{l'})$ {Draft Model forward}
      Compute Loss$(S^{\ell^\star}, E^m, S^L, E^M, T, D)$
      Update $\theta$

---

Table 4 demonstrates that doubling the search space granularity yields negligible performance differences, with maximum speedup variation of only $0.02\times$, confirming that our chosen search space provides adequate coverage of the optimization landscape without requiring computationally expensive fine-grained search. Furthermore, comparing these results with the preliminary experiments in Table 1 reveals STAR's training effectiveness: after full TPPT training, STAR achieves substantially higher speedups ($2.65\times$ vs $2.57\times$ for head pruning configurations, $2.67\times$ vs $2.56\times$ for visual compression) despite maintaining comparable token acceptance lengths. This improvement demonstrates STAR's ability to apply more aggressive pruning strategies during training while preserving draft model quality, enabling superior speed-accuracy trade-offs compared to naive pruning approaches applied without integrated training optimization.

Table 4: Analysis of search space granularity on LLaVA-v1.6-Vicuna-7B at Temperature=0

| Configuration | MMT-Bench | | SEED-Bench-2 | | ScienceQA | | OCRBench | | ChartQA | | MathVista | |
| --- | --- | --- | --- | --- | --- | --- | --- | --- | --- | --- | --- | --- |
| | S | $\tau$ | S | $\tau$ | S | $\tau$ | S | $\tau$ | S | $\tau$ | S | $\tau$ |
| Head Pruning Ratio | 2.65 | 6.32 | 2.54 | 6.20 | 2.44 | 6.10 | 2.02 | 4.84 | 1.92 | 4.35 | 2.03 | 5.22 |
| Double Head Pruning Ratio | 2.66 | 6.34 | 2.53 | 6.18 | 2.44 | 6.09 | 2.01 | 4.82 | 1.90 | 4.38 | 2.03 | 5.19 |
| Vision Compression Ratio | 2.67 | 6.40 | 2.60 | 6.22 | 2.57 | 6.02 | 2.07 | 4.88 | 1.94 | 4.44 | 2.13 | 5.32 |
| Double Vision Compression Ratio | 2.66 | 6.37 | 2.62 | 6.22 | 2.57 | 6.00 | 2.09 | 4.85 | 1.95 | 4.43 | 2.12 | 5.28 |

Table 5 reveals the fundamental trade-off in speculative decoding between draft sequence length and computational efficiency. The draft window size determines the maximum number of tokens the draft model can generate before verification by the target model. As $\gamma$ increases from 4 to 8, the average accepted token length ($\tau$) consistently improves across all benchmarks, with MMT-Bench showing an increase from 4.89 to 7.05 tokens. However, this improvement comes at the cost of reduced speedup ratios, which decline from $2.76\times$ to $2.59\times$ on MMT-Bench. This trade-off occurs because when draft sequences are rejected, the computational cost of generating all the rejected tokens is wasted, and rejection typically happens early in the sequence, rendering most subsequent tokens in longer draft windows wasteful. The diminishing speedup returns beyond $\gamma$=6 suggest an optimal balance point where the computational overhead of generating additional draft tokens begins to outweigh the benefits of potentially longer accepted sequences. This analysis validates our choice of $\gamma$=6 in the main experiments and demonstrates that effective speculative decoding requires careful calibration between speculation aggressiveness and computational efficiency.

Table 5: Evaluation of draft window size ($\gamma$) impact on STAR performance for LLaVA-v1.6-Vicuna-7B at Temperature=0.

| $\gamma$ (Draft Window Size) | MMT-Bench | | SEED-Bench-2 | | ScienceQA | |
|---|---|---|---|---|---|---|
| | S | $\tau$ | S | $\tau$ | S | $\tau$ |
| 4 | 2.76 | 4.89 | 2.68 | 4.79 | 2.53 | 4.87 |
| 5 | 2.73 | 5.78 | 2.69 | 5.62 | 2.56 | 5.09 |
| 6 | 2.67 | 6.27 | 2.61 | 6.18 | 2.45 | 5.71 |
| 7 | 2.62 | 6.72 | 2.55 | 6.54 | 2.41 | 5.93 |
| 8 | 2.59 | 7.05 | 2.51 | 7.07 | 2.36 | 6.37 |

As shown in Figure 4, while entropy captures the average uncertainty of token-level attention distributions, it reflects how dispersed or concentrated the attention is in each layer. However, model stability and information flow depend on how that entropy changes across layers. $\Delta$ entropy highlights fluctuations, revealing whether a layer's attention is becoming more stable or more chaotic relative to its neighbors. If we used only entropy, we would overlook these dynamic shifts and miss layers where abrupt structural changes or information transitions occur. By summing entropy and $\Delta$ entropy, the total metric integrates both the static view of uncertainty and the dynamic view of its variation, providing a more faithful signal for selecting layers and guiding downstream decisions.

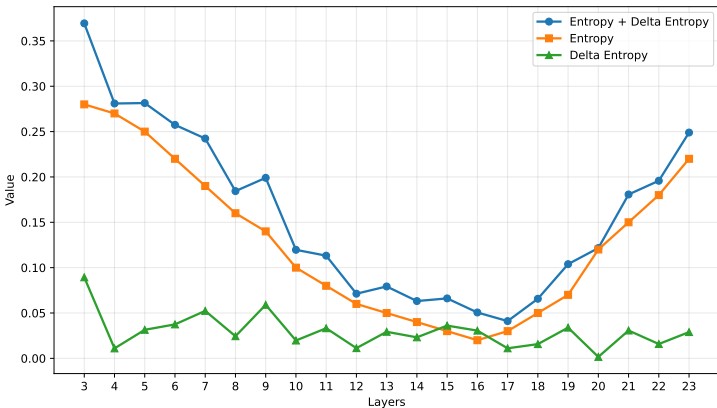

Figure 4: Layer-wise comparison of entropy, $\Delta$entropy, and their sum.

