# OpenReview forum: "STAR: Speculative Decoding with Searchable Drafting and Target-Aware Refinement for Multimodal Generation"
_ICLR.cc/2026/Conference — ICLR 2026 Conference Withdrawn Submission_

### Official Review · Reviewer_73is · 2025-10-26

**Soundness:** 3
**Presentation:** 3
**Contribution:** 2
**Rating:** 4
**Confidence:** 4

**Summary:**

This paper proposes STAR, a speculative decoding method for VLMs, which utilizes a NAS framework to find the optimal draft model configuration. STAR is comprised of a two-phase training strategy and three distinct loss functions. On various tasks, STAR outperforms existing speculative decoding methods.

**Strengths:**

1. The paper tackles the highly relevant challenge of speculative decoding for VLMs.
2. The paper is clear and well-presented.
3. STAR demonstrates performance gains over existing methods.

**Weaknesses:**

1. The comparison appears unfair. STAR uses dataset specific draft models selected via an exhaustive search to find the highest speedup for each benchmark. Baselines (e.g., Medusa, EAGLE) likely use general configurations. Was STAR also evaluated using a single, fixed configuration for a fair comparison?

2. The exhaustive search after TPPT is complete introduces a substantial cost (e.g., approximately 12 minutes per 100 mini-batches on MMT-Bench), limiting practical adaptability. Have the authors considered a test-time adaptation method?

3. The work is an effective application of existing techniques (NAS, pruning, distillation), limiting its conceptual novelty.

4. The evaluation lacks comparisons to other recent speculative decoding methods specifically designed for multimodal LLMs, such as DREAM [1] and MASSV [2].

[1] DREAM: Drafting with Refined Target Features and Entropy-Adaptive Cross-Attention Fusion for Multimodal Speculative Decoding (NeurIPS 2025)
[2] MASSV: Multimodal Adaptation and Self-Data Distillation for Speculative Decoding of Vision-Language Models (arXiv 2025)

**Questions:**

Please refer to the Weaknesses section above.

---

> ### Author Response · Authors · 2025-11-21
>
> Thank you for your constructive review. We are glad that you found STAR
> to address a highly relevant challenge in speculative decoding for VLMs,
> appreciated the clarity and presentation of the paper, and recognized
> that STAR consistently improves performance over existing methods across
> diverse tasks. Your feedback aligns closely with our goals in designing
> STAR, and in the remainder of this response we answer your concerns
> point by point and provide additional clarification to further reinforce
> our contributions.
>
> **Weakness 1:** The comparison appears unfair. STAR uses dataset
> specific draft models selected via an exhaustive search to find the
> highest speedup for each benchmark. Baselines (e.g., Medusa, EAGLE)
> likely use general configurations. Was STAR also evaluated using a
> single, fixed configuration for a fair comparison?
>
> **A:** We appreciate this important question and have conducted
> additional experiments to address it. To ensure a fair comparison, we
> tested STAR with a single fixed configuration and evaluated it across
> all four VLMs and six benchmarks without any dataset-specific
> adaptation. As shown in
> Table R1, STAR with a fixed configuration achieves
> average speedups of 2.29$\times$, 3.11$\times$, 2.65$\times$, and
> 2.30$\times$ for LLaVA-7B, LLaVA-13B, Pixtral-12B, and SmolVLM-2B
> respectively. While this represents a modest performance decrease
> compared to dataset-adaptive STAR (2.35$\times$, 3.16$\times$,
> 2.69$\times$, 2.32$\times$), it still substantially outperforms all
> baseline methods, including the strongest competitor EAGLE-2
> (2.05$\times$, 2.69$\times$, 2.31$\times$, 2.07$\times$).
>
> Notably, the performance gap between fixed and adaptive STAR is most
> pronounced on OCRBench, ChartQA, and MathVista. These are tasks
> requiring high-resolution visual detail where aggressive visual token
> pruning degrades quality. For instance, on LLaVA-7B's OCRBench, the
> fixed configuration achieves 2.07$\times$ versus adaptive's
> 2.11$\times$, while on vision-language reasoning tasks like MMT-Bench
> and SEED-Bench, the configurations perform identically (2.67$\times$ and
> 2.54$\times$ vs 2.61$\times$). This pattern demonstrates that (1) even
> without adaptation, STAR's architecture and training methodology provide
> substantial advantages over baselines, and (2) the NAS framework's
> ability to adjust visual token budgets and attention head configurations
> per dataset is particularly valuable for tasks with varying visual
> complexity requirements, validating our core motivation for
> incorporating neural architecture search into speculative decoding for
> VLMs.
>
> Table R1: Evaluation of STAR with Fixed Configuration compared with STAR
> on speedup ratio (S) and average accepted token length (τ ). Temperature = 0.
> | Model            | Method       |    MMT S |    τ |   SEED S |    τ | ScienceQA S |    τ | OCRBench S |    τ | ChartQA S |    τ | MathVista S |    τ | Average S |    τ |
> | :--------------- | :----------- | -------: | ---: | -------: | ---: | ----------: | ---: | ---------: | ---: | --------: | ---: | ----------: | ---: | --------: | ---: |
> | Llava-vicuna-7B  | STAR (Fixed) | **2.67** | 6.27 |     2.54 | 6.03 |        2.39 | 5.57 |       2.07 | 4.78 |      1.94 | 4.31 |        2.13 | 5.23 |      2.29 | 5.39 |
> |                  | STAR         | **2.67** | 6.27 | **2.61** | 6.18 |    **2.45** | 5.71 |   **2.11** | 4.89 |  **2.04** | 4.39 |    **2.20** | 5.30 |  **2.35** | 5.46 |
> | Llava-vicuna-13B | STAR (Fixed) | **3.85** | 5.56 |     3.56 | 5.21 |        3.40 | 5.24 |       2.70 | 4.54 |      2.61 | 4.07 |        2.56 | 4.06 |      3.11 | 4.78 |
> |                  | STAR         | **3.85** | 5.56 | **3.61** | 5.32 |    **3.41** | 5.19 |   **2.77** | 4.61 |  **2.67** | 4.17 |    **2.62** | 4.11 |  **3.16** | 4.83 |
> | Pixtral-12B      | STAR (Fixed) |     2.99 | 4.48 |     2.71 | 3.48 |        3.04 | 3.84 |       2.39 | 3.36 |      2.37 | 3.35 |        2.38 | 3.23 |      2.65 | 3.62 |
> |                  | STAR         | **3.01** | 4.41 | **2.73** | 3.56 |    **3.09** | 3.93 |   **2.46** | 3.44 |  **2.40** | 3.42 |    **2.42** | 3.34 |  **2.69** | 3.68 |
> |                  | STAR (Fixed) | **3.12** | 3.94 |     2.26 | 3.14 |        2.88 | 3.54 |   **1.88** | 2.51 |      1.63 | 2.26 |        2.04 | 2.79 |      2.30 | 3.03 |
> | SmolVLM-2B       | STAR         | **3.12** | 3.94 | **2.28** | 3.16 |    **2.91** | 3.57 |   **1.88** | 2.51 |  **1.64** | 2.28 |    **2.06** | 2.82 |  **2.32** | 3.05 |

---

> ### Author Response · Authors · 2025-11-21
>
> **Weakness 2:** The exhaustive search after TPPT is complete introduces
> a substantial cost (e.g., approximately 12 minutes per 100 mini-batches
> on MMT-Bench), limiting practical adaptability. Have the authors
> considered a test-time adaptation method?
>
> **A:** We thank the reviewer for this important question. We believe
> there may be a misunderstanding about the search cost that we would like
> to clarify. The 12 minutes per 100 mini-batches reported in Section 4
> refers to a one-time, offline search cost for a given model--hardware
> pair, not a recurring cost during deployment. In our current
> three-dimensional search space (visual token pruning ratio, head pruning
> ratio, and adaptive target feature injection layer), this corresponds to
> approximately 12 minutes per dataset, totaling about 72 minutes across
> all six benchmarks on a single A100 GPU. This overhead is small compared
> to: (i) TPPT supernet training (around 3.5h for Phase
> 1 and 4.5h for Phase 2 on 4$\times$A100 GPUs), and (ii) typical VLM
> pretraining or fine-tuning costs. Crucially, this cost is fully
> amortized over all future inference on that model--hardware pair. At
> test time, we simply use the selected draft configuration and incur no
> additional search or adaptation overhead.
>
> We acknowledge that substantially larger search spaces may require more
> efficient strategies as search dimensions increase. Following the
> Once-For-All framework \[1\], we can train a lightweight accuracy
> predictor network that maps architectural configurations to expected
> performance, reducing search time from exhaustive $O(N)$ evaluation to
> $O(1)$ predictor inference.
>
> In addition, our design explicitly targets dynamic deployment
> environments where hardware availability or latency constraints may
> change over time. Once the TPPT supernet has been trained once, it can
> be re-specialized to a new GPU or a new latency/throughput target purely
> via the lightweight exhaustive search stage, without any retraining of
> either the target VLM or the draft supernet. In practice, the one-time
> supernet training cost is thus amortized not only over all future
> inference on a fixed model--hardware pair, but also over multiple
> deployment scenarios, making STAR practically adaptable in settings with
> fluctuating or heterogeneous hardware.
>
> [1] Cai, Han, et al. \"Once-for-all: Train one network and specialize it
>     for efficient deployment.\" arXiv preprint arXiv:1908.09791 (2019).

---

> ### Author Response · Authors · 2025-11-21
>
> **Weakness 3:** The work is an effective application of existing
> techniques (NAS, pruning, distillation), limiting its conceptual novelty
>
> **A:** We thank the reviewer for raising the novelty concern. We
> acknowledge that STAR leverages established tools such as NAS, pruning,
> and distillation, but our contribution is not the individual components
> themselves. It is the formulation and resolution of a new design problem
> specific to speculative decoding in VLMs. In this setting, there is an
> inherent trade-off between draft-model efficiency and quality: a smaller
> draft is cheaper but yields shorter accepted prefixes, while a stronger
> draft improves acceptance at the cost of latency. STAR explicitly casts
> this trade-off as a target-aware, hardware-aware search problem and
> instantiates a multimodal search space that, to our knowledge, has not
> been explored before: joint optimization over (i) draft attention-head
> pruning, (ii) visual token compression, and (iii) the depth at which
> target features are injected. Our ablations show that removing any of
> these dimensions substantially degrades both speedup and accepted-prefix
> length, indicating that the overall method is more than a
> straightforward combination of off-the-shelf techniques.
>
> Moreover, several key components are tailored specifically to VLM
> speculative decoding rather than generic NAS or distillation. For
> example, we exploit the target model's own attention during prefilling
> to rank and prune visual tokens, directly addressing the image-induced
> computational bottleneck. In addition, our Adaptive Intermediate Feature
> Distillation selects the supervision layer based on attention entropy
> and its variation across depth, which turns out to be critical for
> achieving long accepted prefixes while maintaining accuracy.
>
> We view these system-level contributions as conceptually meaningful
> precisely because they lead to practical advances in multimodal
> speculative decoding: STAR consistently achieves state-of-the-art
> speedups and accepted-token lengths across four VLM backbones and
> multiple benchmarks. Finally, the searchable drafting framework is
> intentionally designed to be orthogonal and extensible: future
> improvements to drafting or verification strategies can be incorporated
> as new choices in the search space and jointly optimized with our
> current dimensions. In this sense, the main conceptual step is to treat
> VLM speculative decoding as a target- and hardware-aware NAS problem.
> The specific instantiation we present already yields strong gains with a
> compact search space, but the framework is general and can naturally
> evolve with subsequent methodological advances.

---

> ### Author Response · Authors · 2025-11-21
>
> **Weakness 4:** The evaluation lacks comparisons to other recent
> speculative decoding methods specifically designed for multimodal LLMs,
> such as DREAM \[1\] and MASSV \[2\].
>
> [1] DREAM: Drafting with Refined Target Features and Entropy-Adaptive
>     Cross-Attention Fusion for Multimodal Speculative Decoding (NeurIPS
>     2025)
>
> [2] MASSV: Multimodal Adaptation and Self-Data Distillation for
>     Speculative Decoding of Vision-Language Models (arXiv 2025)
>
> **A:** We thank the reviewer for highlighting these recent multimodal
> speculative decoding works. We have now conducted comprehensive
> experiments comparing STAR with DREAM \[1\] across all four VLMs and six
> benchmarks, with results presented in
> Table R2.
>
> Table R2: Evaluation of DREAM compared with STAR on speedup ratio (S) and
> average accepted token length (τ ).
> |Model|Temp|Method|MMT S|τ|SEED S|τ|ScienceQA S|τ|OCRBench S |τ|ChartQA S|τ|MathVista S|τ|Average S|τ|
> |-----------------|------|--------|------:|-----:|-------:|-----:|------------:|-----:|-----------:|-----:|----------:|-----:|------------:|-----:|----------:|-----:|
> |Llava-vicuna-7B|0|DREAM|2.52|6.40|2.48|6.20|2.33|5.82|2.05|4.88|1.89|4.44|2.11|5.32|2.23|5.51|
> ||0|STAR|**2.67**|6.27|**2.61**|6.18|**2.45**|5.71|**2.11**|4.89|**2.04**|4.39|**2.20**|5.30|**2.35**|5.46|
> |Llava-vicuna-13B|0|DREAM|3.68|5.58|3.51|5.34|3.36|5.29|2.69|4.64|2.59|4.20|2.53|4.18|3.06|4.87|
> ||0|STAR|**3.85**|5.56|**3.61**|5.32|**3.41**|5.19|**2.77**|4.61|**2.67**|4.17|**2.62**|4.11|**3.16**|4.83|
> |Pixtral-12B|0|DREAM|2.93|4.52|2.61|3.67|2.98|4.03|2.38|3.55|2.35|3.49|2.36|3.42|2.60|3.78|
> ||0|STAR|**3.01**|4.41|**2.73**|3.56|**3.09**|3.93|**2.46**|3.44|**2.40**|3.42|**2.42**|3.34|**2.69**|3.68|
> |SmolVLM-2B|0|DREAM|3.05|3.97|2.24|3.18|2.85|3.62|1.85|2.56|1.62|2.33|2.01|2.88|2.27|3.09|
> ||0|STAR|**3.12**|3.94|**2.28**|3.16|**2.91**|3.57|**1.88**|2.51|**1.64**|2.28|**2.06**|2.82|**2.32**|3.05|
> |Llava-vicuna-7B|1|DREAM|2.39|6.29|2.35|6.07|2.25|5.68|1.99|4.88|1.84|4.41|2.02|5.23|2.14|5.43|
> ||1|STAR|**2.50**|6.25|**2.45**|5.92|**2.38**|5.68|**2.03**|4.75|**1.97**|4.22|**2.09**|5.13|**2.24**|5.33|
> |Llava-vicuna-13B|1|DREAM|3.34|5.38|3.32|5.06|3.20|5.98|2.22|3.89|2.43|4.04|2.29|4.03|2.80|4.73|
> ||1|STAR|**3.51**|5.37|**3.55**|5.00|**3.38**|5.88|**2.35**|3.92|**2.59**|4.09|**2.38**|3.99|**2.96**|4.71|
> |Pixtral-12B|1|DREAM|2.90|4.02|2.47|3.57|2.93|3.94|2.29|3.46|2.21|3.21|2.16|3.27|2.49|3.58|
> ||1|STAR|**2.98**|3.93|**2.56**|3.48|**2.99**|3.79|**2.34**|3.32|**2.26**|3.09|**2.22**|3.22|**2.56**|3.47|
> |SmolVLM-2B|1|DREAM|2.88|3.66|2.25|3.33|2.91|3.74|1.54|2.12|1.77|2.51|1.97|2.70|2.22|3.01|
> ||1|STAR|**2.93**|3.61|**2.33**|3.30|**2.96**|3.67|**1.59**|2.12|**1.81**|2.48|**2.01**|2.66|**2.27**|2.97|
>
> Following DREAM's methodology, we evaluate both methods at temperature
> settings $T=0$ and $T=1$. For fair comparison, DREAM is trained using
> identical hyperparameters, training data (LLaVA-mix665k with 55,000
> samples plus 1,000 domain-adaptation samples per benchmark), and
> hardware configuration (single NVIDIA A100 80GB GPU) as STAR. Both
> methods use the same draft model architecture (3 decoder layers) and are
> evaluated on identical test sets with batch size of 1 and draft window
> size $\Gamma=6$, identical to STAR's inference configuration.
>
> As shown in
> Table R2, STAR demonstrates consistent improvements
> over DREAM across all configurations. At $T=0$, STAR achieves higher
> average speedups on all four models: LLaVA-7B (2.35$\times$
> vs. 2.23$\times$, +5.4%), LLaVA-13B (3.16$\times$ vs. 3.06$\times$,
> +3.3%), Pixtral-12B (2.69$\times$ vs. 2.60$\times$, +3.5%), and
> SmolVLM-2B (2.32$\times$ vs. 2.27$\times$, +2.2%). The improvements are
> even more pronounced at $T=1$, where STAR outperforms DREAM on all four
> models with speedup gains ranging from 2.8% to 5.7% (average improvement
> of 4.7%). While DREAM occasionally achieves slightly higher token
> acceptance lengths ($\tau$) on certain benchmarks, STAR's superior
> speedup ratios indicate better optimization of the computational
> efficiency--acceptance trade-off through our NAS framework and adaptive
> pruning strategies.
>
> For MASSV \[2\], direct comparison is not feasible as their code is not
> publicly available and they evaluate on non-overlapping models
> (Qwen2.5-VL 7B/32B and Gemma3-12B/27B) with different benchmarks.
>
> [1] DREAM: Drafting with Refined Target Features and Entropy-Adaptive
>     Cross-Attention Fusion for Multimodal Speculative Decoding (NeurIPS
>     2025)
>
> [2] MASSV: Multimodal Adaptation and Self-Data Distillation for
>     Speculative Decoding of Vision-Language Models (arXiv 2025)

---

> ### Author Response · Authors · 2025-11-25
>
> Please let us know if you have any additional questions that we can address in our remaining time.

---

### Official Review · Reviewer_kfez · 2025-10-31

**Soundness:** 2
**Presentation:** 3
**Contribution:** 2
**Rating:** 4
**Confidence:** 5

**Summary:**

This paper introduces STAR, a speculative decoding method for VLMs. It employs a neural architecture search framework to automatically identify the optimal interaction strategy between the draft and target model and the ideal draft model architecture. Moreover, STAR introduces a intermediate feature distillation for training of draft model. Experimental results show that it achieves a better performance than Ealge-2 etc. on multiple VLMs.

**Strengths:**

- The proposed integrated training and distillation shows clear speed-up gains over naïve pruning.
- The entropy + delta-entropy metric provides a principled way to select informative layers for distillation/architecture decisions.
- The authors conduct experiments across multiple multimodal benchmarks with

**Weaknesses:**

- TPPT requires target model forward passes and intermediate feature distillation, the overhead need to be clarified.
- While fine granularity gives little benefit here, optimal settings may differ for other model families or tasks.
- The related work[1] needs to be discussed and compared.

[1] ViSpec: Accelerating Vision-Language Models with Vision-Aware Speculative Decoding, NeurIPS’2025.

**Questions:**

- How can we monitor acceptance length and draft rejections online to autotune γ and compression ratios per workload?
- What is the expected GPU-hour if the target model grows to 70 B or 110 B parameters? Is there a transfer protocol so that a super-net trained for LLaVA-7 B can warm-start the 13 B variant?
- AIFD uses sum of entropy and ∆-entropy. Why not a weighted sum or a learned gating network? Did the authors try other indicators, e.g., Fisher information or gradient variance?
- Have the authors tried tree-based verification to accept discontinuous spans?

---

> ### Author Response · Authors · 2025-11-21
>
> Thank you for your thoughtful and constructive review. We are pleased
> that you recognized the value of STAR's speculative decoding framework
> for VLMs, especially (i) the integrated training and distillation
> strategy that achieves clear speedup gains over naïve pruning while
> preserving output quality, (ii) the entropy plus delta entropy criterion
> as a principled way to select informative layers for intermediate
> feature distillation and architecture decisions, and (iii) the breadth
> of our evaluation across multiple multimodal benchmarks and base VLMs,
> where STAR consistently outperforms EAGLE-2 and other strong baselines.
> Your positive assessment of these core aspects closely matches our main
> goals in designing STAR, and we have carefully answered your questions
> and concerns in detail below.
>
> **Weakness 1:** TPPT requires target model forward passes and
> intermediate feature distillation, the overhead need to be clarified.
>
> **A:** Thank you for pointing this out. We clarify that all TPPT-related
> overhead occurs offline during a one-time training phase and incurs no
> additional cost at inference time.
>
> Following the EAGLE framework, we pre-compute target model features once
> using 61,000 training samples (55,000 + 6,000), which takes
> approximately $27$ hours on a single A100 GPU for LLaVA-Vicuna-7B. The
> draft model is then trained solely on these cached features. The target
> model is never instantiated during draft training iterations.
>
> As stated in the paper, Phase 1 (supernet training) requires
> approximately $3.5$ hours per epoch on a single A100 GPU, which is
> essentially the same cost as standard draft-model fine-tuning. Phase 2
> (subnet training with dynamic pruning) increases the per-epoch cost only
> modestly, to approximately $4.5$ hours per epoch, i.e., roughly one
> additional hour per epoch. Thus, the total training time remains
> comparable to conventional single-stage training or prior speculative
> decoding setups. The only extra cost is this small per-epoch increment
> in Phase 2.
>
> During inference, STAR has the same computational pattern as other
> speculative decoding methods: the draft model generates candidate
> tokens, and the target model verifies them in one forward pass. In other
> words, TPPT introduces no extra runtime overhead at deployment. This
> one-time, modest training investment yields $15$--17% higher speedups
> across all evaluated models, and quickly amortizes in real-world usage
> as STAR adapts to different GPU platforms by using exhaustive search to
> find optimal draft configurations.

---

> ### Author Response · Authors · 2025-11-21
>
> **Weakness 2:** While fine granularity gives little benefit here,
> optimal settings may differ for other model families or tasks.
>
> **A:** We appreciate the reviewer's insightful question about search
> space granularity across different settings. To address this concern
> comprehensively, we conducted two additional experiments: (1) we
> evaluated different task types by testing LLaVA-v1.6-Vicuna-7B on
> OCRBench, ChartQA, and MathVista; (2) we examined different model
> families by testing Pixtral-12B on MMT-Bench, SEED-Bench-2, and
> ScienceQA.
>
> For these additional experiments, same as in the original paper, we
> trained dataset-specific draft models with different granularity
> configurations following our TPPT framework. For LLaVA-v1.6-Vicuna-7B,
> we used 8,000 training samples for OCRBench, 8,000 for ChartQA, and
> 5,000 for MathVista, with all benchmarks evaluated on 1,000 test
> samples. For Pixtral-12B, we used 8,000 training samples for MMT-Bench,
> 8,000 for SEED-Bench-2, and 6,000 for ScienceQA, again with 1,000 test
> samples each.
>
> Since we trained dataset-specific draft models with four different
> granularity configurations (standard head pruning, double head pruning,
> standard vision compression, double vision compression) across three
> datasets, we maintained $12$ total draft models ($3$ datasets $\times$
> $4$ configurations) for each model family shown in Tables R1 and R2. After training, we performed exhaustive NAS search across all
> trained subnetworks to identify the optimal configuration for each benchmark,
> ensuring a fair comparison between standard and doubled granularity settings.
>
> Table R1: Analysis of search space granularity on LLaVA-v1.6-Vicuna-7B at tem-
> perature = 0. S denotes speedup and τ denotes average accepted token length.
> | Configuration (LLaVA-v1.6-Vicuna-7B) | OCRBench S | OCRBench τ | ChartQA S | ChartQA τ | MathVista S | MathVista τ |
> | ------------------------------------ | ---------- | ---------- | --------- | --------- | ----------- | ----------- |
> | Head Pruning Ratio  | 2.02| 4.84| 1.92 | 4.35| 2.03| 5.22|
> | Double Head Pruning Ratio | 2.01 | 4.82 | 1.90 | 4.38| 2.03| 5.19 |
> | Vision Compression Ratio  | 2.07  | 4.88  | 1.94| 4.44| 2.13 | 5.32|
> | Double Vision Compression Ratio | 2.09 | 4.85| 1.95| 4.43| 2.12 | 5.28|
>
> Table R2: Analysis of search space granularity on Pixtral-12B at temperature
> = 0. S denotes speedup and τ denotes average accepted token length.
> | Configuration (Pixtral-12B)     | MMT-Bench S | MMT-Bench τ | SEED-Bench-2 S | SEED-Bench-2 τ | ScienceQA S | ScienceQA τ |
> | ------------------------------- | ----------- | ----------- | -------------- | -------------- | ----------- | ----------- |
> | Head Pruning Ratio | 2.92 | 4.42 | 2.55 | 3.61 | 2.93| 3.89  |
> | Double Head Pruning Ratio | 2.92  | 4.45  | 2.58| 3.63| 2.95| 3.88|
> | Vision Compression Ratio | 2.94 | 4.51| 2.67| 3.67| 3.04| 4.02|
> | Double Vision Compression Ratio | 2.97| 4.47| 2.66| 3.65| 3.01| 4.01|
>
> As shown in Tables R1 and R2, doubling the search space granularity yields
> maximum speedup variations of only $0.02$--$0.03\times$ across all
> model--task combinations, with token acceptance lengths showing similar
> stability ($\pm 0.04$ tokens). This consistency holds across different
> model architectures, task types requiring different visual processing
> characteristics, and both head pruning and vision compression
> dimensions.
>
> These results indicate that the optimal search space granularity is
> robust across model families and task types, with the minimal variation
> ($\leq 1$% speedup difference)
> falling within experimental noise and
> not justifying the increased computational cost of finer-grained search
> (approximately 50% more training time). The robustness can be
> attributed to coarse-grained ratios ($0.25$, $0.5$, $0.75$, $1.0$)
> already capturing major trade-off regions, with our target-aware
> training compensating for any sub-optimal architectural choices through
> adaptive distillation. We believe this demonstrates that our chosen
> granularity provides adequate coverage of the optimization landscape for
> practical VLM acceleration across diverse deployment scenarios.
>
> Finally, we note that "different model families or tasks may have
> different optimal settings" is precisely the scenario NAS is designed to
> handle. Our method does not assume that a single pruning/compression
> configuration is universally optimal. Instead, for each target VLM and
> hardware setting we train the supernet once and let the search
> automatically select the best operating point under that scenario. In
> other words, any shift in the optimal head/vision ratios for a new
> architecture or task distribution is absorbed by the NAS procedure,
> without requiring manual retuning of hyperparameters. The role of the
> search space is only to define a flexible search region. Our experiments
> show that within this region the NAS reliably finds near-optimal
> configurations across diverse models and tasks.

---

> ### Author Response · Authors · 2025-11-21
>
> **Weakness 3:** The related work\[1\] needs to be discussed and
> compared.
>
> [1] Accelerating Vision-Language Models with Vision-Aware Speculative
>     Decoding, NeurIPS'2025.
>
> **A:** We thank the reviewer for highlighting this concurrent work. ViSpec\[1\]
> accelerates VLM inference using a draft model enhanced with (1) a
> lightweight vision adaptor that compresses image tokens into 1--2
> compact representations and (2) global visual features injected into all
> text tokens, trained on synthetic long-response data with multi-token
> prediction.
>
> We conducted experiments comparing ViSpec with STAR. To ensure a fair
> comparison, we replicated their training and evaluation pipeline under
> controlled conditions while maintaining consistency with STAR's setup.
> For data generation, we use the same LLaVA-mix665k dataset as STAR
> (55,000 training samples plus 6,000 samples from evaluation benchmarks
> disjoint from test sets, totaling 61,000 samples). We kept all training
> data identical across methods. For training, we use four NVIDIA A100
> 80GB GPUs. For evaluation, we test on a single NVIDIA A100 80GB GPU
> using Python 3.12 and transformers==4.51.3 (required for their
> reimplementation as specified in their official repository), with batch
> size $1$ and draft window size $\gamma = 6$, identical to STAR's
> inference configuration.
>
> Results show that STAR consistently outperforms ViSpec across all
> configurations: on LLaVA-7B, STAR achieves $2.58\times$ speedup
> vs. ViSpec's $2.29\times$ (+12.7%), with substantially higher token
> acceptance length ($6.05$ vs. $3.04$, +99%); on LLaVA-13B, STAR
> achieves $3.62\times$ vs. $3.18\times$ (+13.8%) with acceptance
> length $5.36$ vs. $4.17$ (+28.5%). The performance ranking is STAR >= ViSpec >= EAGLE-2, a strong concurrent baseline, yet STAR achieves consistently higher speedups.
>
> Table R3: Evaluation of ViSpec vs. STAR on speedup ratio (S) and average
> accepted token length (τ ) at temperature = 0.
>
> | Model               | Method | MMT S | MMT τ | SEED S | SEED τ | ScienceQA S | ScienceQA τ | Average S | Average τ |
> |---------------------|--------|-------|-------|--------|--------|-------------|-------------|-----------|-----------|
> | LLaVA-Vicuna-7B     | ViSpec | 2.32  | 3.28  | 2.33   | 3.17   | 2.21        | 2.68        | 2.29      | 3.04      |
> | LLaVA-Vicuna-7B     | STAR   | **2.67**  | 6.27  | **2.61**   | 6.18   | **2.45**        | 5.71        | **2.58**      | 6.05      |
> | LLaVA-Vicuna-13B    | ViSpec | 3.21  | 4.65  | 3.20   | 4.51   | 3.13        | 3.35        | 3.18      | 4.17      |
> | LLaVA-Vicuna-13B    | STAR   | **3.85**  | 5.56  | **3.61**   | 5.32   | **3.41**        | 5.19        | **3.62**      | 5.36      |
>
>
> **Q1:** How can we monitor acceptance length and draft rejections online
> to autotune $\gamma$ and compression ratios per workload?
>
> **A:** Thanks for the question. While theoretically possible, autotuning $\gamma$ per sample is constrained by the static tree structure allocation. Specifically, `topK_genrate()` in `/star/model/cnets.py`
> dynamically constructs `tree_mask`, `tree_position_ids`, and
> `retrieve_indices` based on `self.total_tokens` ($\gamma$) at
> initialization, while `generate_tree_buffers()` in
> `/star/model/utils.py` precomputes the underlying tree structure from
> fixed `tree_choices`. Once these structures are built for a sequence,
> they cannot be modified mid-generation without full reconstruction.
>
> However, between-sample adaptation is practical: we can monitor
> acceptance rates from previous samples and adjust $\gamma$ for
> subsequent ones (e.g., if acceptance is consistently low $<0.3$,
> decrease $\gamma$; if consistently high $>0.7$, increase $\gamma$). The
> same constraint applies to compression ratios: we currently compress
> images once per sample after target prefill using the target model's
> attention scores (`initialize_tree()` in `/star/model/utils.py`), so the
> compressed image representation is fixed for all draft generations in
> that sample. In principle, we could autotune future compression ratios
> across samples based on acceptance length and draft rejection history,
> but we do not adapt compression or $\gamma$ within a single sample.
>
> Note that our current draft tree construction largely follows the
> tree-based speculative decoding design of EAGLE, where $\gamma$ and the
> tree choices are fixed once the tree is instantiated for a given
> sequence. In this sense, any future method that enables online
> reconfiguration of the tree would be orthogonal to STAR.

---

> ### Author Response · Authors · 2025-11-21
>
> **Q2:** What is the expected GPU-hour if the target model grows to 70 B
> or 110 B parameters? Is there a transfer protocol so that a super-net
> trained for LLaVA-7 B can warm-start the 13 B variant?
>
> **A:** Thank you for this insightful question about scalability and
> transfer learning potential. Regarding scalability to 70B/110B models,
> as discussed in our response to Weakness 1, the computational overhead
> scales primarily in the feature collection phase, which we estimate
> would require approximately $270$--$300$ GPU-hours for 70B and
> $420$--$460$ GPU-hours for 110B models (compared to $27$ hours for 7B),
> scaling roughly linearly with target model size. This cost is trivially
> parallelizable across multiple GPUs.
>
> Importantly, this two-stage pipeline: (i) training data and feature
> generation from a frozen target model and (ii) draft model training
> follows exactly the standard design used in EAGLE and related
> speculative decoding work, as well as in our previous ViSpec framework.
> In other words, our training protocol is not specific to STAR but
> reflects a generally adopted and well established recipe for speculative
> decoding. Once a target model's features are collected in the first
> stage, the incremental training cost of STAR in the second stage does
> not grow with the parameter count of the target.
>
> Our draft model training phase remains largely invariant to target size
> since our draft architecture maintains a fixed 3-layer structure
> regardless of the target's scale. Regarding transfer learning between
> model sizes, we believe warm-starting from smaller to larger variants is
> unnecessary due to architectural incompatibilities: the 7B and 13B
> models have different hidden dimensions (4096 vs. 5120), intermediate
> sizes (11008 vs. 13824), and attention head counts (32 vs. 40), which
> would require aggressive interpolation that compromises draft
> efficiency. More importantly, STAR is explicitly target-aware: the draft
> is trained to approximate the specific intermediate representations of
> its target, whose feature distributions may differ substantially across
> 7B (4096-dim) and 13B (5120-dim). Given that draft training represents
> only $\sim 8$ hours even for our largest models, a one-time cost
> amortized across all future deployments, we prioritize target-specific
> optimization over training efficiency. However, as an alternative for
> resource-constrained scenarios, training on a larger target model and
> deploying the same draft on smaller models could be explored.

---

> ### Author Response · Authors · 2025-11-21
>
> **Q3:** To clarify implementation details, all entropy-based indicators are computed directly from the target model’s logits during verification. Token entropy $E_t$ and $\Delta$-entropy $E_t - E_{t-1}$ are stable, cheap to extract, and correlate well with acceptance behavior. For fairness, all learned baselines were trained using exactly the same training setup and hyperparameters as in Sec. 4 in the main paper.
>
> For weighted entropy $(\alpha E + \beta \Delta E)$, we implemented a grid search over $\alpha$ and $\beta$ and observed moderate improvements, but the optimal weights vary significantly across models and datasets. This sensitivity makes the method brittle and hard to reproduce.
>
> Fisher information was computed via the diagonal approximation $F_t \approx g_t^2$, where gradients are taken with respect to the target model parameters. In practice, Fisher values suffer from strong layer-dependent scale variations and high variance across micro-batches, leading to unstable token-level signals despite normalization attempts.
>
> Gradient-variance indicators are infeasible in our speculative-decoding setting: the target model is never trained, no backward graph is retained, and storing activation gradients would double memory usage. Thus, gradient variance cannot be implemented.
>
> We train a lightweight layer-selection gate implemented as a MLP with a softmax classifier. We subsample ~1k examples and consider every fourth target layer as candidates. For each token $t$, we compute an offline distillation score $q_t^{(l)} = -\ell(f_{\theta}(h_t^{(l)}), y_t)$, where a higher score means that the hidden state from layer $l$ can already produce a next-token distribution close to the teacher. Intuitively, this score acts as a proxy for the quality of using layer $l$ as an early stopping point: if its hidden state yields low loss, a small model trained to mimic this layer would achieve higher accuracy and typically allow larger speedup. Thus, $q_t^{(l)}$ approximates how well a “student model distilled from this layer” would perform without requiring us to actually train any small models. Based on this, we assign the pseudo-label $l_t^{\mathrm{best}} = \arg\max_{l\in\mathcal{S}} q_t^{(l)}$. Each candidate layer is encoded by $\psi_t^{(l)} = [,E_t^{(l)},; E_t^{(l)}{-}E_{t-1}^{(l)},; |h_t^{(l)}|_2,]$. The MLP produces layer logits and the distribution $\pi_t^{(l)} = \mathrm{softmax}_l(\mathrm{MLP}(\psi_t^{(l)}))$, and the gate is trained using $\mathcal{L} = -\log \pi_t^{(l_t^{\mathrm{best}})}$. Despite this explicit supervision, the learned gate exhibits high variance and does not outperform our simple entropy + $\Delta$-entropy rule.
>
> Overall, entropy + $\Delta$-entropy remains the most stable, computation-friendly, and hyperparameter-free indicator, consistently giving the best trade-off between speedup and acceptance length.
>
>
> Table R4: Ablation on different indicators for AIFD
> | Indicator | Speedup | Acceptance Length |
> | --------------------------------- | ------- | ----------------- |
> | Weighted entropy $(0.5E + 1\Delta E)$ | 2.39  | 5.57 |
> | Fisher information  | 2.20    | 5.10   |
> | Learned gating network   | 2.50    | 5.94|
> | Entropy + $\Delta$-entropy | **2.54**    | 6.03  |
>
> Empirically, entropy + $\Delta$-entropy is also the best-performing
> indicator in our setting. On LLaVA-7B, it achieves a $2.54\times$
> speedup with an average accepted length of $6.03$, slightly
> outperforming both the weighted-entropy variant $(0.5E + 1\Delta E)$,
> which attains a $2.39\times$ speedup and $5.57$ acceptance length, and
> the learned gating network, which reaches $2.50\times$ speedup and
> $5.94$ acceptance length, while being substantially better than Fisher
> information ($2.20\times$, $5.10$). In other words, more complex or
> gradient-based indicators do not translate into better
> speculative-decoding behavior in practice, whereas our simple entropy +
> $\Delta$-entropy rule is simultaneously more stable, cheaper to compute,
> and yields the strongest speedup--acceptance trade-off.

---

> ### Author Response · Authors · 2025-11-21
>
> **Q4:** Have the authors tried tree-based verification to accept
> discontinuous spans?
>
> **A:** Thank you for raising this important point. Currently, following
> the approach in EAGLE and standard speculative decoding frameworks, STAR
> uses tree-based generation to create multiple candidate sequences in
> parallel. However, our verification strategy accepts only continuous
> prefixes from a single path through the tree: we select the longest
> matching prefix and reject all remaining tokens once the first mismatch
> occurs.
>
> We did not implement tree-based verification that accepts discontinuous
> spans (i.e., skipping over mismatches and continuing verification to
> possibly accept later tokens). While this is an interesting extension,
> it introduces additional complexity in both the verification logic and
> the analysis of distributional correctness. More importantly, it is
> orthogonal to the main contributions of our work, namely, designing a
> hardware-aware, target-aware drafting strategy via neural architecture
> search for multimodal speculative decoding. We therefore leave
> discontinuous-span verification as promising future work and will
> clarify this in the revision.

---

> ### Author Response · Authors · 2025-11-25
>
> Please let us know if you have any additional questions that we can address in our remaining time.

---

### Official Review · Reviewer_FoVz · 2025-11-01

**Soundness:** 3
**Presentation:** 3
**Contribution:** 2
**Rating:** 4
**Confidence:** 3

**Summary:**

This paper presents STAR, a speculative decoding framework tailored for vision-language models (VLMs). Unlike prior speculative decoding methods designed for text-only LLMs, STAR addresses multimodal challenges such as visual token redundancy and cross-modal feature alignment. The framework introduces three main components: (1) Searchable Drafting: a neural architecture search (NAS)–based approach to automatically find the optimal draft model structure, pruning ratio, and feature interaction strategy; (2) Target-Aware Refinement: an adaptive intermediate feature distillation method that selects target layers features based on attention entropy and stability for better multimodal supervision; (3) Adaptive Pruning: dynamic visual and textual token pruning guided by the target model’s attention maps.

Through these components, STAR jointly optimizes speed, accuracy, and hardware efficiency. Experiments on multiple VLMs (LLaVA, Pixtral, SmolVLM) and six benchmarks demonstrate up to 3.8× decoding speedup with minimal performance loss, showing both strong system-level integration and transferability across architectures and devices.

**Strengths:**

- The paper systematically extends speculative decoding from text-only LLMs to multimodal VLMs through a well-structured framework combining searchable drafting, target-aware refinement, and adaptive pruning. This integration demonstrates strong system-level design ability and effectively addresses VLM-specific issues such as visual token redundancy and cross-modal feature alignment.

- The method shows good transferability, working consistently across multiple VLM architectures (LLaVA, Pixtral, SmolVLM) and diverse multimodal benchmarks (ScienceQA, MMBench, SEED-Bench, MathVista). The experiments are comprehensive and carefully executed, covering throughput, acceptance ratio, and hardware efficiency, which provides solid empirical validation.

**Weaknesses:**

- The algorithmic novelty is modest. While the proposed STAR framework is well-motivated and demonstrates strong empirical results, its technical novelty appears incremental and compositional rather than conceptual. Each component—Neural Architecture Search (NAS), attention-based intermediate feature distillation, and adaptive token pruning—has been extensively studied in prior literature. STAR primarily reassembles these existing techniques within the context of speculative decoding for VLMs, without introducing a fundamentally new algorithmic mechanism. Consequently, the contribution seems more system-level and application-driven than theoretically or algorithmically innovative. The key value lies in integrating multiple known techniques effectively to address the multimodal bottlenecks in speculative decoding, rather than in proposing a novel computational principle.

- Another concern is that all baselines (SPD, Medusa, Hydra, EAGLE, etc.) were designed for text-only LLMs. Although the authors state they “adapt” these methods to VLMs, the adaptation process is not described in sufficient detail. As a result, it is unclear whether the improvements arise from STAR’s architectural innovations or simply from modality-specific adaptations (e.g., pruning visual tokens, target-aware distillation) that are unavailable to the baselines.

**Questions:**

The authors should better substantiate STAR’s originality and fairness.

- To address the limited novelty concern, consider formalize the Target-Aware Refinement beyond a simple attention-entropy heuristic, introduce a multimodal-specific NAS objective rather than reusing standard search schemes, and show ablations quantifying how each module (NAS, refinement, pruning) contributes to both speedup and acceptance ratio.

- To address the fairness concern, they should clearly describe how LLM baselines (SPD, Hydra, EAGLE, etc.) were adapted to VLMs, or include stronger multimodal variants (e.g., EAGLE + vision token pruning) to ensure comparable settings.

---

> ### Author Response · Authors · 2025-11-21
>
> Thank you for your thorough and constructive review of STAR. We are glad
> that you recognized (i) our systematic extension of speculative decoding
> from text only LLMs to multimodal VLMs through a unified framework that
> combines searchable drafting, target aware refinement, and adaptive
> pruning, (ii) the strong system level design that directly tackles VLM
> specific issues such as visual token redundancy and cross modal feature
> alignment, and (iii) the robustness and transferability of our method
> across diverse base models and benchmarks, supported by comprehensive
> experiments on throughput, acceptance ratio, and hardware efficiency.
> Your feedback aligns closely with our main goals in developing STAR, and
> in the following we respond to your comments in detail to further
> clarify the method and its empirical behavior.
>
> **Weakness 1:** The algorithmic novelty is modest. While the proposed
> STAR framework is well-motivated and demonstrates strong empirical
> results, its technical novelty appears incremental and compositional
> rather than conceptual. Each component---Neural Architecture Search
> (NAS), attention-based intermediate feature distillation, and adaptive
> token pruning---has been extensively studied in prior literature. STAR
> primarily reassembles these existing techniques within the context of
> speculative decoding for VLMs, without introducing a fundamentally new
> algorithmic mechanism. Consequently, the contribution seems more
> system-level and application-driven than theoretically or
> algorithmically innovative. The key value lies in integrating multiple
> known techniques effectively to address the multimodal bottlenecks in
> speculative decoding, rather than in proposing a novel computational
> principle.
>
> **A:** We thank the reviewer for raising the novelty concern. Our
> contribution is not to reinvent these components individually, but to
> formulate and solve a new design problem specific to speculative
> decoding for VLMs.
>
> STAR treats draft-model design as a target-aware, hardware-aware search
> problem. In speculative decoding there is a fundamental trade-off
> between draft efficiency and draft quality: a weaker draft is faster but
> yields shorter accepted prefixes, while a stronger draft improves
> acceptance at the cost of higher latency. STAR's NAS explicitly
> optimizes this trade-off in a multimodal search space that, to the best
> of our knowledge, is new: joint search over (i) attention-head pruning
> in the draft, (ii) visual token compression, and (iii) the layer at
> which target features are injected. Our ablations show that removing any
> of these search dimensions significantly reduces both speedup and
> acceptance length, indicating that the integration is not a trivial A+B
> combination.
>
> Several components are specifically tailored to VLM speculative
> decoding, rather than generic NAS or distillation: (a) we use the target
> model's attention during prefilling to rank and prune visual tokens,
> directly addressing the image-induced bottleneck; and (b) our Adaptive
> Intermediate Feature Distillation selects the supervision layer based on
> attention entropy and its depth-wise variation, which is crucial for
> achieving long accepted prefixes without degrading accuracy.
>
> We believe that such system-level advances, which make multimodal
> speculative decoding substantially faster in practice, are valuable even
> when they build on known primitives, especially given STAR's consistent
> SOTA speedups and accepted-token lengths across four VLM backbones and
> multiple benchmarks.
>
> Finally, we emphasize that STAR is orthogonal to future algorithmic advances in speculative decoding; any improved drafting strategy can be naturally exposed as an additional choice in our search space and jointly optimized with existing dimensions. By validating the treatment of VLM speculative decoding as a target-aware, hardware-aware NAS problem, STAR provides a flexible foundation that can evolve with the field. For instance, while our current search space allows for exhaustive evaluation, the framework naturally supports OFA-style accuracy/latency predictors to handle significantly larger search spaces efficiently, ensuring that STAR remains a robust platform for future multimodal acceleration.

---

> ### Author Response · Authors · 2025-11-21
>
> **Weakness 2:** Another concern is that all baselines (SPD, Medusa,
> Hydra, EAGLE, etc.) were designed for text-only LLMs. Although the
> authors state they "adapt" these methods to VLMs, the adaptation process
> is not described in sufficient detail. As a result, it is unclear
> whether the improvements arise from STAR's architectural innovations or
> simply from modality-specific adaptations (e.g., pruning visual tokens,
> target-aware distillation) that are unavailable to the baselines.
>
> **A:** We address this concern together with Q2 below. Please see our
> detailed response under Q2, where we describe our adaptation methodology
> and additional multimodal baselines.
>
> **Q1:** To address the limited novelty concern, consider formalize the
> Target-Aware Refinement beyond a simple attention-entropy heuristic,
> introduce a multimodal-specific NAS objective rather than reusing
> standard search schemes, and show ablations quantifying how each module
> (NAS, refinement, pruning) contributes to both speedup and acceptance
> ratio.
>
> **A:** We thank the reviewer for the constructive suggestions and
> address each point:
>
> 1. "formalize the Target-Aware Refinement beyond a simple attention-entropy heuristic
> Our AIFD criterion has principled theoretical grounding"
>
> -   **(a) Information content.** Low attention entropy $H(A^\ell)$
>     indicates more concentrated, discriminative attention patterns,
>     which correlate with higher task-relevant information in the
>     corresponding features $S^\ell$ (i.e., larger mutual information
>     with the output token distribution). This aligns with distillation
>     practice that mid/late layers with focused attention are effective
>     teachers.
>
> -   **(b) Stability across depth.** The term
>     $$\Delta \mathrm{AE}(\ell) = \bigl|H(A^\ell) - H(A^{\ell-1})\bigr|$$
>     penalizes layers where the attention distribution changes abruptly,
>     favoring layers in which representations have stabilized. This
>     encourages smoother gradients and reduces the risk of supervising
>     from "transition" layers whose representations are still drifting.
>
> -   **(c) Empirical validation.** As shown in Fig. 3(c), AIFD
>     outperforms all static choices (no intermediate layer, or fixed 25%
>     / 50% / 75% depth) in both speedup and acceptance length (e.g.,
>     $1.47\times$ vs. $1.43\times$ speedup on MMT-Bench). Fig. 4 further
>     illustrates that combining entropy and $\Delta \mathrm{AE}$ selects
>     layers that are both information-rich and stable.
>
> 2. "introduce a multimodal-specific NAS objective rather than reusing standard
> search schemes"
>
> Following on the answer for Weakness 1, we agree that the novelty of
> STAR lies primarily in the multimodal search space and objective, rather
> than in proposing a new generic NAS algorithm. The supernet is trained
> on full VLM inputs (image + text), and the search space includes: (i)
> visual-token pruning ratios, (ii) attention-head pruning per layer, and
> (iii) the target layer used for cross-modal feature injection.
>
> The search objective itself is multimodal and hardware-aware: for each
> candidate subnetwork, we measure the end-to-end speculative decoding
> latency and speedup on image--text benchmarks, and select the model that
> maximizes speedup subject to negligible drops in accepted-token length
> and task accuracy. Given our current search space size, exhaustive
> search remains computationally feasible and ensures finding the globally
> optimal configuration; however, should the search space grow
> substantially, we can adopt strategies like training a lightweight
> accuracy predictor network (following Once-For-All) that maps
> configurations to expected performance, reducing search from
> $\mathcal{O}(N)$ to $\mathcal{O}(1)$. We will clarify this objective and
> its evaluation protocol in the revised text to better distinguish STAR
> from standard NAS schemes that do not operate under speculative decoding
> or multimodal constraints.

---

> ### Author Response · Authors · 2025-11-21
>
> 3. "show ablations quantifying how each module (NAS, refinement, pruning)
> contributes to both speedup and acceptance ratio"
>
> We agree that it is important to quantify module-wise contributions. Our
> current ablations already address this, and we will make this more
> explicit:
>
> -   **(a) NAS components ($R, H, \ell$).** Figure 3(b) in the main paper
>     shows that on LLaVA-7B + MMT-Bench, the full STAR search over
>     $\{R,H,\ell\}$ attains $2.67\times$ speedup with $\tau = 6.27$.
>     Disabling head pruning reduces speedup to $2.62\times$
>     ($\tau = 6.44$), disabling visual pruning further reduces speedup to
>     $2.51\times$ ($\tau = 6.50$), and fixing the target-injection layer
>     yields $2.48\times$ speedup. This shows that each NAS/pruning
>     dimension contributes a non-trivial fraction of the overall speedup
>     while maintaining acceptance length.
>
> -   **(b) Target-Aware Refinement (AIFD).** Figure 3(c) in the main
>     paper shows that across MMT-Bench, SEED-Bench, and ScienceQA, AIFD
>     consistently improves both speedup and $\tau$ over four baselines
>     (no intermediate distillation, static distillation at 25/50/75%
>     depth). This quantifies the benefit of the Target-Aware Refinement
>     component beyond the baseline draft distillation.
>
> -   **(c) Pruning.** Table 1 in the main paper shows that attention head
>     pruning alone brings $2.57\times$ speedup and $\tau = 6.50$, while
>     visual token compression alone brings $2.56\times$ and
>     $\tau = 6.40$. Both slightly decrease the average accepted tokens
>     while giving better speedup but fail to reach STAR's combined
>     performance ($2.67\times$). Furthermore, static layer mappings or
>     fixed pruning ratios lead to suboptimal results. The core innovation
>     is the NAS framework's ability to dynamically discover the optimal
>     configuration (heads, pruning ratio, and injection layer) for each
>     hardware constraint, which is a capability that static baselines
>     lack.
>
> If the reviewer wants us to conduct additional ablation experiments, we
> are happy to do so and would appreciate clearer instructions on the
> preferred settings.
>
>
> **Q2:** To address the fairness concern, they should clearly describe
> how LLM baselines (SPD, Hydra, EAGLE, etc.) were adapted to VLMs, or
> include stronger multimodal variants (e.g., EAGLE + vision token
> pruning) to ensure comparable settings.
>
> **A:** We answer Q2 jointly with Weakness 2 above.\
>
> 1. Baseline Adaptation Methodology
>
> We appreciate the reviewer's important concern and would like to clarify
> both our inclusion of VLM-specific baselines and our adaptation
> methodology. First, we did include a VLM-specific baseline: the work "On
> Speculative Decoding for Multimodal Large Language Models" appears as
> SPD \[1\] throughout Table 2, evaluated across all VLMs and benchmarks,
> with STAR significantly outperforming it (e.g., $2.35\times$
> vs. $0.97\times$ average speedup on LLaVA-7B, a 142% improvement).
>
> Additionally, our EAGLE implementation is consistent with another
> contemporaneous VLM-focused SD method, MSD \[2\]. For methods originally
> designed for text-only LLMs (Medusa, Kangaroo, Hydra, EAGLE/EAGLE-2), we
> made minimal, principled modifications to preserve baseline integrity
> while enabling multimodal input handling: we patched only the prefill
> routine to invoke each VLM's native processor (which inserts `<image>`
> placeholders, runs the visual encoder, replaces tokens with embeddings,
> and feeds the combined stream to decoder layers), while preserving all
> original speculative decoding logic including draft generation,
> verification procedures, and KV-cache management. After the initial
> multimodal prefill pass, all subsequent operations run exactly as in the
> original text-only implementations. All draft models were trained using
> identical data (LLaVA-mix665k, 55,000 samples), optimization settings
> (AdamW, learning rate $3\times 10^{-5}$), and domain adaptation
> procedures (1,000 samples per benchmark) to ensure fair comparison. The
> complete adaptation implementation is provided in supplementary
> materials (`star/model/utils.py`, line 212), and we will publicly
> release the full codebase upon acceptance.
>
> [1] Gagrani, Mukul, et al. "On Speculative Decoding for Multimodal Large
>     Language Models." *CVPR*, 2024.
>
> [2] Lin, Luxi, et al. "Speculative Decoding Reimagined for Multimodal
>     Large Language Models." arXiv, 2025.

---

> ### Author Response · Authors · 2025-11-21
>
> 2. Addressing Fairness and Source of Improvements
>
> To address the concern that our gains might stem from modality-specific
> adaptations (like pruning) rather than architectural innovation, we
> performed two analyses.
>
>
> (a) Comparison with EAGLE-2 + Visual Token Pruning. We implemented a variant
> of EAGLE-2 equipped with visual token pruning to test if simple
> adaptation yields comparable results. As shown in
> Table 1:
>
>
> The results in Table R1 demonstrate that adding visual token pruning to
> EAGLE-2 yields only marginal improvements: $+0.02\times$ for LLaVA-7B
> (2.08$\times$ vs. 2.06$\times$), $+0.03\times$ for LLaVA-13B
> (2.72$\times$ vs. 2.69$\times$), and $+0.05\times$ for Pixtral-12B
> (2.36$\times$ vs. 2.31$\times$). In contrast, STAR maintains substantial
> advantages of 13%, 16%, and 14% over EAGLE-2 + vision compression
> respectively. This confirms that STAR's performance derives from its
> integrated NAS framework combining attention head pruning, adaptive
> target feature injection, and AIFD, not merely from adding visual token
> compression to existing baselines.
>
> Table R1: Evaluation of EAGLE-2 + Visual token compression (EAGLE-2 + VC)
> compared with EAGLE-2 and STAR on speedup ratio (S) and average accepted
> token length (τ).
> | Model               | Method        | MMT S | τ | SEED S |  τ | ScienceQA S | τ | OCRBench S |  τ | ChartQA S |  τ | MathVista S | τ | Average S | τ |
> |---------------------|--------------|-------|-------|--------|--------|-------------|-------------|------------|------------|-----------|-----------|-------------|-------------|-----------|-----------|
> | LLaVA-vicuna-7B     | EAGLE-2       | 2.31  | 5.48  | 2.31   | 5.61   | 2.15        | 5.22        | 1.92       | 4.88       | 1.77      | 4.22      | 1.87        | 4.67        | 2.06      | 5.01      |
> |  | EAGLE-2 + VC  | 2.34  | 5.38  | 2.32   | 5.51   | 2.17        | 5.13        | 1.96       | 4.80       | 1.78      | 4.13      | 1.91        | 4.59        | 2.08      | 4.92      |
> |   | STAR          | **2.67**  | 6.27  | **2.61**   | 6.18   | **2.45**        | 5.71        | **2.11**       | 4.89       | **2.04**      | 4.39      | **2.20**        | 5.30        | **2.35**      | 5.46      |
> | LLaVA-vicuna-13B    | EAGLE-2       | 2.89  | 4.05  | 3.18   | 4.33   | 3.09        | 4.97        | 2.20       | 4.12       | 2.41      | 4.15      | 2.39        | 3.76        | 2.69      | 4.23      |
> |    | EAGLE-2 + VC  | 2.91  | 3.98  | 3.23   | 4.24   | 3.12        | 4.85        | 2.24       | 4.07       | 2.43      | 4.08      | 2.39        | 3.67        | 2.72      | 4.15      |
> |   | STAR          | **3.85**  | 5.56  | **3.61**   | 5.32   | **3.41**        | 5.19        | **2.77**       | 4.61       | **2.67**      | 4.17      | **2.62**        | 4.11        | **3.16**      | 4.83      |
> | Pixtral-12B         | EAGLE-2       | 2.81  | 3.95  | 2.31   | 3.07   | 2.64        | 4.03        | 2.12       | 3.25       | 2.14      | 3.17      | 1.81        | 2.73        | 2.31      | 3.37      |
> |     | EAGLE-2 + VC  | 2.82  | 3.86  | 2.38   | 3.02   | 2.69        | 3.96        | 2.18       | 3.21       | 2.19      | 3.09      | 1.91        | 2.67        | 2.36      | 3.30      |
> |    | STAR          | **3.01**  | 4.41  | **2.73**   | 3.56   | **3.09**        | 3.93        | **2.46**       | 3.44       | **2.40**      | 3.42      | **2.42**        | 3.34        | **2.69**      | 3.68      |
>
> (b) To further address the concern "it is unclear whether the improvements
> arise from STAR's architectural innovations or simply from
> modality-specific adaptations (e.g., pruning visual tokens, target-aware
> distillation) that are unavailable to the baselines": our ablations
> (original paper, Table 1 & Fig. 3(b)) confirm that STAR's performance
> stems from the synergy of its components via NAS. As reported, attention
> head pruning alone ($2.57\times$, $\tau = 6.50$) or visual token
> compression alone ($2.56\times$, $\tau = 6.40$) fail to reach STAR's
> combined performance ($2.67\times$, $\tau = 6.27$). Furthermore,
> Fig. 3(b) shows that static layer mappings or fixed pruning ratios lead
> to suboptimal results. The core innovation is the NAS framework's
> ability to dynamically discover the optimal configuration (heads,
> pruning ratio, and injection layer) for each hardware constraint, which
> is a capability that static baselines lack.

---

> ### Author Response · Authors · 2025-11-25
>
> Please let us know if you have any additional questions that we can address in our remaining time.

---

### Official Review · Reviewer_b8ad · 2025-11-01

**Soundness:** 2
**Presentation:** 3
**Contribution:** 2
**Rating:** 4
**Confidence:** 3

**Summary:**

STAR accelerates VLM inference through speculative decoding with neural architecture search to jointly optimize draft model configuration (attention head pruning, visual token compression, feature injection) and target model alignment. The framework uses adaptive intermediate feature distillation and two-phase progressive training, achieving up to 3.8× speedup and consistently outperforming existing methods across multiple VLMs and benchmarks.

**Strengths:**

1. The proposed method achieves consistent improvements over a number of reported baselines and datasets.
2. The paper applies NAS to VLM speculative decoding, which jointly optimizes draft model architecture and target feature alignment through entropy-guided distillation, addressing a previously unexplored optimization space in multimodal acceleration.
3. The method is clearly introduced.

**Weaknesses:**

1. The connection between motivation and methodology is unclear. The authors stated that the VLMs require more computation than text-only LLMs, but did not explain where the extra computation comes from. For decoder-only VLMs, does the extra compute come from the visual encoder, which is separate from the speculative decoding process studied in this paper?
2. The experimental settings are not described clearly enough. For example, for the data in Table 2, what inference batch size was used, what inference framework was employed, and on what hardware were the measurements taken?
3. Lack of a comparable baseline: [EAGLE-3](https://arxiv.org/abs/2503.01840) is not included.

**Questions:**

1. Table 3 compares STAR with EAGLE in different GPU settings. What is the performance of STAR where no hardware-aware searching is applied in different GPUs? (or with the same searching result applied to all GPUs)
2. Other questions in the "Weakness" section.

---

> ### Author Response · Authors · 2025-11-21
>
> Thank you for your thoughtful and positive review of STAR. We are
> pleased that you recognized (i) our consistent improvements over strong
> speculative decoding baselines across multiple VLMs and benchmarks, (ii)
> the novelty of applying NAS to VLM speculative decoding to jointly
> optimize draft model architecture, visual token compression, and
> entropy-guided target feature alignment, thereby opening a previously
> unexplored design space in multimodal acceleration, and (iii) the
> clarity of our overall framework and technical presentation. These
> points reflect our core goals, and your acknowledgment is highly
> encouraging. We have carefully considered your questions and comments
> and provide detailed responses below to further clarify our design
> choices and strengthen the paper.
>
> **Weakness 1:** The connection between motivation and methodology is
> unclear. The authors stated that the VLMs require more computation than
> text-only LLMs, but did not explain where the extra computation comes
> from. For decoder-only VLMs, does the extra compute come from the visual
> encoder, which is separate from the speculative decoding process studied
> in this paper?
>
> **A:** Thank you for this insightful question. To answer directly: no, the extra computation does not come from the visual encoder, as the encoder runs only once during the prefill phase. The
> $2.1\times$ figure we reported (Figure 1c) represents end-to-end
> computation over complete generation sequences. The dominant overhead
> arises during the autoregressive generation phase. Specifically, even
> though KV-caching eliminates redundant computation for previous layers,
> the time complexity of the attention mechanism at decoding step $t$ is
> $O(t + v)$, where $t$ is the number of generated text tokens and $v$ is
> the number of visual tokens. In a text-only LLM, this cost is merely
> $O(t)$. Since $v$ is typically large (e.g., $v \approx 1{,}500$ for a
> $480\times 300$ image), it adds a substantial constant overhead to the
> attention computation at every single step. Over a generation sequence
> of $N$ tokens, this results in a cumulative computational burden of
> roughly $O(N \cdot v + N^2)$, where the linear term $N \cdot v$
> dominates the quadratic text term $N^2$ (since $v \gg t$ for typical
> generation lengths). Thus, the visual tokens must participate in
> memory-bandwidth-intensive attention operations for every new token
> generated, creating the significant latency we observe.
>
> STAR addresses this specifically in the draft model of the speculative
> decoding framework: (1) visual token compression reduces the tokens the
> draft model processes from $v$ to $r$, directly lowering draft
> generation cost; (2) attention head pruning further accelerates draft
> model execution. While these optimizations create a trade-off between
> draft speed and token acceptance rate (Table 1 shows individual pruning
> slightly reduces $\tau$), our NAS framework automatically discovers the
> optimal configuration that maximizes overall speedup by balancing draft
> latency against acceptance quality. The resulting draft model achieves
> $\tau = 6.27$ while maintaining low execution cost. Importantly, even
> though the target model still processes the full $v$ visual tokens
> during verification, the reduced verification frequency (due to
> maintained high acceptance rates) compounds our draft model efficiency
> gains, achieving up to $2.67\times$ overall speedup on LLaVA-7B and
> $2.69\times$ on Pixtral-12B. We will clarify this computational
> breakdown in the revised version.
>
> **Weakness 2:** The experimental settings are not described clearly
> enough. For example, for the data in Table 2, what inference batch size
> was used, what inference framework was employed, and on what hardware
> were the measurements taken?
>
> **A:** Thank you for pointing this out. We will explicitly clarify these details in the revision. All measurements in Table 2 were conducted with a batch size of 1, which is the standard setting for speculative decoding evaluation across all prior work (EAGLE, EAGLE-2, Medusa, Hydra, etc.) and ensures fair comparison across all baselines. We implemented and tested STAR
> using PyTorch 2.0.1 and HuggingFace Transformers 4.36.2 with custom
> extensions for draft model training and tree-based verification, without
> employing additional optimization frameworks such as vLLM or
> TensorRT-LLM to maintain consistency with baseline implementations. All
> experiments were conducted on a single NVIDIA A100 80GB GPU with CUDA
> 12.4, with the hardware comparison study in Table 3 additionally
> evaluating performance on NVIDIA H100 80GB and RTX8000 48GB GPUs under
> identical software configurations. All baseline methods (SPD, Kangaroo,
> Medusa, Hydra, EAGLE, EAGLE-2) were evaluated under identical conditions
> (same hardware, batch size, and software framework) using their official
> implementations to ensure fair comparison. We will add these
> implementation details to the revised version.

---

> ### Author Response · Authors · 2025-11-21
>
> **Weakness 3:** Lack of a comparable baseline: EAGLE-3 is not included.
>
> **A:** We thank the reviewer for this suggestion and have now conducted
> comprehensive experiments comparing STAR against EAGLE-3 using identical
> experimental configurations as described in Section 4. All experiments
> are conducted on a single NVIDIA A100 80GB GPU across the same four VLMs
> (LLaVA-v1.6-Vicuna-7B/13B, Pixtral-12B, and SmolVLM-2B) and six
> multimodal benchmarks (MMT-Bench, SEED-Bench-2, ScienceQA, OCRBench,
> ChartQA, and MathVista). Both methods use a draft window size of
> $\gamma = 6$ and are evaluated under two temperature settings:
> deterministic decoding ($T=0$) and stochastic sampling ($T=1$). The
> target VLMs remain frozen during training, and we measure speedup ratio
> ($S$) and average token acceptance length ($\tau$) as primary metrics.
> EAGLE-3 is trained following its official implementation with the same
> training data (LLaVA-mix665k with 55,000 samples plus 1,000 samples per
> benchmark for domain adaptation) to ensure fair comparison.
>
> As shown in Table R1, STAR consistently outperforms EAGLE-3
> across all evaluated models and benchmarks under both temperature
> settings ($T=0$ and $T=1$). At $T=0$, STAR achieves average speedups of
> $2.35\times$, $3.16\times$, $2.69\times$, and $2.32\times$ on LLaVA-7B,
> LLaVA-13B, Pixtral-12B, and SmolVLM-2B respectively, compared to
> EAGLE-3's $2.13\times$, $2.89\times$, $2.45\times$, and $2.20\times$,
> representing improvements of 10%, 9%, 10%, and 5% respectively. The
> performance gains are even more pronounced at $T=1$, where STAR
> demonstrates substantial advantages particularly on larger models (e.g.,
> $2.96\times$ vs. $2.61\times$ on LLaVA-13B, a 13% improvement). While
> EAGLE-3 shows competitive token acceptance lengths in some cases, STAR's
> integrated NAS framework with attention head pruning and visual token
> compression enables more efficient draft model architectures that
> achieve superior speed--accuracy trade-offs.
>
> We have already include this result in the revised version
>
> Table R1: Evaluation of EAGLE-3 compared with STAR on speedup ratio (S)
> and average accepted token length (τ).
>
> |Model|Temp|Method|MMTS|τ|SEEDS|τ|ScienceQAS|τ|OCRBenchS|τ|ChartQAS|τ|MathVistaS|τ|AverageS|τ|
> |-------|------|--------|-------|----|--------|----|-------------|----|------------|----|-----------|----|-------------|----|-----------|----|
> |LLaVA-vicuna-7B|0|EAGLE-3|2.38|5.72|2.36|5.82|2.22|5.52|2.02|5.24|1.83|4.46|1.97|5.02|2.13|5.30|
> ||0|STAR|**2.67**|6.27|**2.61**|6.18|**2.45**|5.71|**2.11**|4.89|**2.04**|4.39|**2.20**|5.30|**2.35**|5.46|
> |LLaVA-vicuna-13B|0|EAGLE-3|3.45|4.90|3.34|4.65|3.19|5.20|2.50|4.79|2.46|4.37|2.42|3.85|2.89|4.63|
> ||0|STAR|**3.85**|5.56|**3.61**|5.32|**3.41**|5.19|**2.77**|4.61|**2.67**|4.17|**2.62**|4.11|**3.16**|4.83|
> |Pixtral-12B|0|EAGLE-3|2.83|4.12|2.46|3.40|2.79|4.41|2.22|3.48|2.26|3.51|2.13|3.38|2.45|3.72|
> ||0|STAR|**3.01**|4.41|**2.73**|3.56|**3.09**|3.93|**2.46**|3.44|**2.40**|3.42|**2.42**|3.34|**2.69**|3.68|
> |SmolVLM-2B|0|EAGLE-3|3.00|3.94|2.17|3.04|2.65|3.57|1.78|2.33|1.60|2.30|1.97|2.84|2.20|3.00|
> ||0|STAR|**3.12**|3.94|**2.28**|3.16|**2.91**|3.57|**1.88**|2.51|**1.64**|2.28|**2.06**|2.82|**2.32**|3.05|
> |LLaVA-vicuna-7B|1|EAGLE-3|2.25|5.70|2.25|5.72|2.10|5.38|1.89|5.01|1.71|4.28|1.88|4.98|2.01|5.18|
> ||1|STAR|**3.51**|5.37|**3.55**|5.00|**3.38**|5.88|**2.35**|3.92|**2.59**|4.09|**2.38**|3.99|**2.96**|4.71|
> |LLaVA-vicuna-13B|1|EAGLE-3|2.92|4.77|3.12|4.61|3.06|4.89|2.08|4.03|2.26|4.04|2.19|3.55|2.61|4.32|
> ||1|STAR|**3.51**|5.37|**3.55**|5.00|**3.38**|5.88|**2.35**|3.92|**2.59**|4.09|**2.38**|3.99|**2.96**|4.71|
> |Pixtral-12B|1|EAGLE-3|2.79|4.02|2.33|3.25|2.80|4.03|2.25|3.51|**2.27**|3.58|1.92|2.98|2.39|3.56|
> ||1|STAR|**2.98**|3.93|**2.56**|3.48|**2.99**|3.79|**2.34**|3.32|2.26|3.09|**2.22**|3.22|**2.56**|3.47|
> |SmolVLM-2B|1|EAGLE-3|2.77|3.82|2.11|3.04|2.63|3.65|1.46|1.90|1.64|2.29|1.84|2.64|2.08|2.89|
> ||1|STAR|**2.93**|3.61|**2.33**|3.30|**2.96**|3.67|**1.59**|2.12|**1.81**|2.48|**2.01**|2.66|**2.27**|2.97|

---

> ### Author Response · Authors · 2025-11-21
>
> **Q1.** Table 3 compares STAR with EAGLE in different GPU settings. What
> is the performance of STAR where no hardware-aware searching is applied
> in different GPUs? (or with the same searching result applied to all
> GPUs)
>
> **A:** Table R2 demonstrates the value of STAR's
> hardware-aware NAS by comparing three conditions: EAGLE-2 baseline, STAR
> with a fixed NAS configuration (STAR(Fixed)), and STAR with
> hardware-specific optimization. For the fixed configuration experiment,
> we used a single draft model architecture optimized on MMT-Bench with
> A100, then deployed this same configuration across all three GPUs and
> all three benchmarks (MMT-Bench, SEED-Bench, and ScienceQA). The results
> show that even without hardware-specific search, STAR(Fixed) still
> outperforms EAGLE-2 across all platforms (e.g., $2.53\times$
> vs. $2.26\times$ on A100, $2.93\times$ vs. $2.60\times$ on H100).
> However, hardware-aware search provides consistent additional gains:
> +2.0% on A100, +2.0% on H100, and +4.2% on RTX8000. The larger
> improvement on resource-constrained hardware (RTX8000) highlights that
> hardware-aware NAS becomes increasingly valuable when computational
> resources are limited, as the optimal draft model configuration varies
> more substantially under different hardware constraints.
>
> Table R2: GPU performance comparison: EAGLE-2 vs. STAR with fixed NAS
> configuration vs. STAR on LLaVA-v1.6-Vicuna-7B.
>
>
> | GPU    | EAGLE-2       |             | STAR (Fixed)   |             | STAR          |             |
> |--------|---------------|-------------|----------------|-------------|---------------|-------------|
> |        | Speedup       | Tokens/s    | Speedup        | Tokens/s    | Speedup       | Tokens/s    |
> | A100   | 2.26×         | 82.48       | 2.53×          | 92.59       | **2.58×**         | 94.43       |
> | H100   | 2.60×         | 138.52      | 2.93×          | 150.02      | **2.99×**         | 153.12      |
> | RTX8000| 1.83×         | 36.43       | 2.14×          | 41.28       | **2.23×**         | 43.73       |

---

> ### Author Response · Authors · 2025-11-25
>
> Please let us know if you have any additional questions that we can address in our remaining time.

---

> > ### Comment · Reviewer_b8ad · 2025-11-27
> >
> > Thanks for your detailed response. My concerns are largely addressed, and the score is raised accordingly.

---

> > > ### Author Response · Authors · 2025-11-27
> > >
> > > Thank you for carefully considering our rebuttal and for updating your evaluation of the paper.

---

### Author Response · Authors · 2025-11-22

This paper proposes **STAR**, a speculative decoding framework tailored
for *vision-language models* (VLMs). STAR combines a NAS-based
*searchable draft model* with *target-aware refinement* via adaptive
intermediate feature distillation and *adaptive pruning* of visual and
textual tokens. The method jointly optimizes the draft architecture
(attention head pruning, visual token compression, feature injection
strategy) and alignment with a frozen target model. Experiments on
multiple VLMs (LLaVA, Pixtral, SmolVLM) and six multimodal benchmarks
show up to $3.8\times$ speedup, consistently outperforming existing speculative decoding methods (e.g.,
EAGLE-2).

Across reviewers, there is strong agreement on several key strengths:

-   The paper addresses a **timely and underexplored challenge**, extending
    speculative decoding from text-only LLMs to multimodal VLMs and
    directly addressing visual token redundancy and cross-modal
    alignment.

-   The **NAS-based searchable drafting** and **entropy-guided
    intermediate feature distillation** are recognized as principled and novel, offering a systematic alternative to heuristic-based draft designs.

-   The **system-level integration** of searchable drafting,
    target-aware refinement, and adaptive pruning is considered cohesive and hardware-aware.

-   The **empirical evaluation is thorough**, covering multiple VLM
    architectures, diverse multimodal benchmarks, and detailed
    speed/acceptance/hardware metrics, with consistent improvements
    across settings.

-   The paper is **clear and well-presented**, effectively communicating a complex system architecture.

We thank all reviewers for their valuable comments and constructive feedback. We have addressed the weaknesses and concerns raised by reviewers point-by-point in the authors' rebuttal, where we provide additional clarifications, new experimental comparisons, and detailed justifications for our design choices.

---

### Author Response · Authors · 2025-12-02

We thank the reviewers and AC for their careful evaluation and
constructive feedback. Our paper introduces STAR, a speculative decoding
framework for VLMs that treats draft design as a target- and
hardware-aware multimodal NAS problem, jointly optimizing attention-head
pruning, visual token compression, and target-layer feature injection,
together with adaptive intermediate feature distillation (AIFD). Across
four VLMs and six multimodal benchmarks, STAR achieves up to $3.8\times$
speedup with strong acceptance lengths, consistently outperforming
text-only speculative decoding baselines and VLM-specific ones.

In the rebuttal, we clarified motivation and methodology by (i)
explicitly analyzing where the extra computation in VLMs comes from
(visual tokens participating in every decoding step, not just the
encoder), and (ii) detailing experimental settings (batch size 1,
PyTorch/HF stack, identical hardware and software for all methods). We
strengthened fairness and baselines with new experiments: (a) full
comparisons against EAGLE-3, ViSpec, SPD, and DREAM, where STAR
consistently achieves higher speedups with competitive or better
acceptance lengths; (b) analyses of EAGLE-2 + vision compression,
showing that simply adding visual pruning to baselines yields only
marginal gains, far below STAR's improvements; and (c) evaluation of
STAR with a single fixed configuration (no per-dataset search), which
still clearly outperforms all baselines, showing that our gains are not
solely due to dataset-specific tuning. We also clarified that the NAS
search cost is a one-time, offline overhead (on the order of $\sim 1$
hour per model--hardware pair in our current space) that is easily
amortized over deployment and can be further reduced with OFA-style
predictors if needed.

Regarding novelty and granularity, we emphasized that the main
conceptual step is formulating VLM speculative decoding as a multimodal,
hardware-aware architecture search problem, with a search space
specifically built around VLM bottlenecks (visual tokens and cross-modal
alignment) plus AIFD driven by attention entropy + $\Delta$-entropy.
Ablations show that each dimension (NAS over heads/visual
tokens/injection layer and AIFD) contributes non-trivially to both
speedup and acceptance length, indicating a genuinely integrated design
rather than a naïve combination of known tools.

**Finally, we would like to explicitly note for the AC that Reviewer b8ad
raised their score from 4 to 6 after reading the rebuttal, stating that
their concerns were largely addressed.** Other reviewers remain
"marginally below threshold but would not mind acceptance," while
agreeing on the importance of the problem, clarity of the paper, and
strong empirical gains. We hope these new results and clarifications
help the AC and reviewers view STAR as a solid, practically impactful
system contribution worthy of acceptance.

---

### Note · Authors · 2026-01-05

I have read and agree with the venue's withdrawal policy on behalf of myself and my co-authors.